# Domain-Separated Quantum Neural Network for Truss Structural Analysis with Mechanics-Informed Constraints

**DOI:** 10.3390/biomimetics10060407

**Published:** 2025-06-16

**Authors:** Hyeonju Ha, Sudeok Shon, Seungjae Lee

**Affiliations:** School of Industrial Design & Architectural Engineering, Korea University of Technology & Education, 1600 Chungjeol-ro, Byeongcheon-myeon, Cheonan 31253, Republic of Korea; hayezii@koreatech.ac.kr (H.H.); leeseung@koreatech.ac.kr (S.L.)

**Keywords:** quantum neural network (QNN), variational quantum circuit (VQC), truss system, force method, entanglement, quantum gates, surrogate model

## Abstract

This study proposes an index-based quantum neural network (QNN) model, built upon a variational quantum circuit (VQC), as a surrogate framework for the static analysis of truss structures. Unlike coordinate-based models, the proposed QNN uses discrete member and node indices as inputs, and it adopts a separate-domain strategy that partitions the structure for parallel training. This architecture reflects the way nature organizes and optimizes complex systems, thereby enhancing both flexibility and scalability. Independent quantum circuits are assigned to each separate domain, and a mechanics-informed loss function based on the force method is formulated within a Lagrangian dual framework to embed physical constraints directly into the training process. As a result, the model achieves high prediction accuracy and fast convergence, even under complex structural conditions with relatively few parameters. Numerical experiments on 2D and 3D truss structures show that the QNN reduces the number of parameters by up to 64% compared to conventional neural networks, while achieving higher accuracy. Even within the same QNN architecture, the separate-domain approach outperforms the single-domain model with a 6.25% reduction in parameters. The proposed index-based QNN model has demonstrated practical applicability for structural analysis and shows strong potential as a quantum-based numerical analysis tool for future applications in building structure optimization and broader engineering domains.

## 1. Introduction

Truss structures are widely used across various fields, such as architecture, civil engineering, mechanical engineering, and aerospace engineering. These structural systems are composed of straight members that are connected at joints and offer advantages, such as a high strength-to-weight ratio, efficient use of materials, and stable load distribution. In particular, each truss member primarily carries axial force with minimal influence from moments. Although their geometry is discrete, space trusses exhibit mechanical behavior similar to that of continua, making their analysis and prediction of physical behavior critically important in design, damage assessment, and maintenance [1,2]. In general, the analysis of truss structures relies on methods such as the finite element method (FEM) or the force method [2]. While these methods offer high accuracy, they incur significant computational cost when applied to large-scale systems and face limitations in capturing complex nonlinear behavior. Tasks such as dynamic analysis [2], structural stability evaluation [3], and various forms of optimization (e.g., topology, shape, and weight) [1,4,5], as well as shape control [6,7], require appropriate analysis techniques. Consequently, there is growing demand for efficient and reliable advanced analysis methods and surrogate modeling approaches. Recently, the advancement of artificial neural networks (ANNs) and the development of diverse training strategies have led to their active application in structural mechanics. ANNs have been used as surrogate models for nonlinear analysis, stability evaluation, and truss optimization, and they have also served as experimental platforms for validating novel neural network architectures. Due to their powerful ability to approximate complex functions and solve high-dimensional nonlinear problems, neural networks have been widely adopted in structural analysis [8,9,10,11]. Physics-Informed Neural Networks (PINNs), which integrate physical laws directly into the training process, have emerged as a promising approach for effectively solving structural analysis problems [12]. However, conventional neural network models often require large amounts of training data and substantial computational resources to achieve optimal generalization performance.

The implementation of quantum computing technologies has been proposed as a potential alternative to overcome these limitations. In particular, with the advent of the Noisy Intermediate-Scale Quantum (NISQ) era, various quantum algorithms have been actively investigated to transcend the boundaries of classical approaches. Among them, variational quantum algorithms (VQAs) have attracted significant attention as a methodology that solves optimization problems using quantum circuits, offering the possibility of greater expressivity with fewer parameters than conventional artificial neural networks [13]. While several heuristic-based quantum algorithms have been proposed for optimization tasks [5], circuit-based quantum approaches have shown considerable potential in a variety of engineering applications. In particular, quantum neural networks (QNNs), a subclass of VQAs, leverage quantum properties, such as entanglement and superposition, to effectively approximate high-dimensional data spaces, making them viable candidates for surrogate modeling in structural engineering problems. However, to date, no research has been reported on applying circuit-based quantum models to discrete structural systems, such as trusses. In this study, all QNN implementations were conducted on a simulator, taking into account the current limitations of quantum hardware. This represents an exploratory phase aimed at assessing the future applicability of actual quantum devices.

This study proposes a quantum neural network (QNN)-based framework for solving structural analysis problems in truss structures. Conventional neural network-based models for structural analysis primarily rely on coordinate-based inputs and often fail to adequately reflect the discrete characteristics inherent to truss systems. The proposed QNN framework instead utilizes an index-based input representation, where identifiable discrete information, such as member and node numbers, is used as input rather than spatial coordinates. This design aligns with the structural nature of trusses, where spatial interpolation is meaningless and only nodal responses carry physical significance. In particular, this study maps a single index to multiple structural features (e.g., members, nodes, and loading conditions), and it applies domain separation techniques to ensure computational flexibility and scalability in QNN training. Based on this understanding, we newly apply a QNN framework that leverages quantum entanglement and superposition to truss structure analysis, presenting new possibilities for tackling high-dimensional structural problems. Moreover, a mechanics-informed loss function based on the force method is constructed to integrate physical reliability into QNN training. This research is the first to incorporate circuit-based QNNs into the structural analysis of trusses by combining discrete index inputs with residual-based, physics-informed loss functions, thereby offering a modeling approach distinct from traditional FEM and classical neural networks. This attempt not only serves as an empirical assessment of QNN potential and performance in structural analysis, but also lays the groundwork for applying quantum machine learning to engineering problems, even when under current quantum hardware limitations. Practically, it suggests potential expansion into a novel computational framework for large-scale structural systems, while theoretically contributing to the understanding of QNN expressivity and the effectiveness of Lagrangian-duality-based loss formulations.

The main contributions of this study are as follows:Proposal of a QNN framework for truss structure analysis: A quantum circuit-based QNN model was developed to analyze truss structures, demonstrating new possibilities for application.Design of index-based input and domain decomposition: Instead of using coordinates, the model uses unique indices of members and nodes as inputs, and it applies domain decomposition to ensure flexibility and scalability.Implementation of a mechanics-informed loss function: Physical constraints based on the force and displacement methods were integrated into a Lagrangian dual framework to construct the loss function.Numerical validation: The predictive performance of the QNN was verified through 2D and 3D truss examples, and the effects of the qubit count and layer depth were analyzed.

The remainder of this paper is organized as follows. Section 2 presents the theoretical foundation for applying QNNs to structural optimization problems based on recent advances in quantum-based methods with a review of relevant prior research. Section 3 introduces the force method widely used in the analytical approach for truss structures and formulates the structural analysis problem applicable to QNNs by mathematically defining the stiffness representation, degree of freedom decomposition, and external–internal force relationships. Section 4 describes the components of the VQA framework and outlines the proposed QNN. It also defines an example problem and constructs the total loss function by incorporating the residual, penalty, and Lagrangian terms. In Section 5, the performance of the proposed QNN model is evaluated using a series of numerical experiments. This analysis includes not only prediction accuracy metrics, such as mean squared error (MSE; $L_{2}$ error) and the coefficient of determination ($R^{2}$), but also convergence characteristics, the number of qubits, circuit depth, and sensitivity to individual loss components. Additionally, Section 5.4 provides a comprehensive evaluation of the proposed QNN model’s performance in a structural and comparative analysis with previous studies based on the numerical results obtained in this study. Finally, Section 6 summarizes the proposed research and results, focusing on the applicability of QNNs to structural analysis and potential advancements in quantum circuit optimization.

## 2. Preliminary

A truss structure is generally modeled as a system of one-dimensional elastic members that transmit only axial forces and are connected at nodes. Each member is governed by a boundary value problem (BVP) with essential and natural boundary conditions defined at its ends. Although individual members are treated as discrete systems, the overall truss structure requires a global formulation that ensures compatibility and equilibrium at the connecting joints. Among the various methods that satisfy these conditions, the finite element method (FEM) is the most widely used, ultimately leading to a system of equations based on the global stiffness matrix K.

Although the finite element method (FEM) provides high accuracy in structural analysis, its numerical stability and computational complexity increase significantly as the system size grows or when extended to high-dimensional and nonlinear problems. In particular, for pin-jointed or flexible structures, geometric nonlinearity arising from shape changes requires more efficient analytical methods. Furthermore, the stiffness matrix is assembled to satisfy the boundary value problems of individual elements, and the internal forces are indirectly computed from the global displacement field of the system. The force method was introduced as an alternative to address these limitations, and it has been continuously developed by Linkwitz and Schek (1971) [14], Schek (1974) [15], Tibert and Pellegrino (2003) [16], and Saeed (2014, 2016) [17,18]. Unlike the finite element method, the direct control method converts deformed shapes obtained through shape analysis into control actions using the principles of the force method. This approach has been highlighted for its advantages in providing intuitive control results. The force method was developed in parallel with the advancement of structural shape control techniques. Shape control was first introduced in 1984 by Weeks [19], who demonstrated its feasibility for antenna structures using a finite element model. In 1985, Haftka and Adelman [20] proposed a static shape control procedure for flexible structures by incorporating temperature-based control elements. Subsequent studies by Edberg (1987) [21], Burdisso and Haftka (1990) [22], and Sener (1994) [23] introduced control strategies based on adjusting member lengths using actuators. Precision shape control using piezoelectric actuators has been actively investigated since 1999 and in the early 2000s. Since the mid-2000s, shape control techniques have been expanded to incorporate intelligent control strategies, such as probabilistic search methods, genetic algorithms, and induced strain actuation theory. Korkmaz (2011) [24] classified these control strategies into active control, adaptive control, and intelligent control. Meanwhile, shape control using the force method gained momentum with the study by Kwan and Pellegrino in 1994 [25], who investigated the placement and operation of control members in space structures. In 1997, You [26] proposed a displacement control technique for cable structures based on the modification of the member lengths. In 2007, Dong and Yuan [27] validated the potential of shape adjustment using pre-stressed members. Kawaguchi (1996) [28] addressed the constant control problem, which involves the simultaneous control of displacements and internal forces. Subsequent studies by Xu and Luo (2009) [29], Wang (2013) [30], and Saeed (2014) [17] have further advanced the development of constant control techniques. The direct method that is based on the force method is computationally efficient. However, it has limitations in clearly defining the optimal state and requires selecting a single solution among many possibilities. To address this issue, various indirect optimization methods have been proposed. Nevertheless, certain limitations persist even when applying structural analysis, optimization, or control using the FEM or force method. Various algorithms have been introduced to overcome these challenges, including genetic algorithms, gradient-based methods, and dimensionality reduction techniques.

In recent years, quantum computing has emerged as an alternative method for solving complex control and optimization problems. In contrast to classical computing, quantum computing employs qubits as the fundamental unit of information processing. A qubit can exist in a superposition of basis states 0 and 1 and is generally represented as follows:(1)$$|\psi \rangle =\alpha |0\rangle +\beta |1\rangle ,\;\alpha ,\beta \in {C,\;|\alpha |}^{2}+{\left|\beta \right|}^{2}=1.$$

The measurement causes the quantum state to collapse probabilistically into one of the basis states. Computation can be performed using quantum circuits that combine entanglement between qubits and quantum gates. These fundamental properties of quantum systems offer computational potential for efficiently exploring high-dimensional and nonlinear design spaces, which are often intractable for classical optimization methods. Based on this potential, variational quantum algorithms (VQAs) have recently emerged as promising techniques for structural analyses and optimization problems. Traditional quantum algorithms include the HHL (Harrow–Hassidim–Lloyd (HHL) algorithm for solving linear systems, Grover’s algorithm for unstructured search problems, and quantum phase estimation (QPE) for precise eigenvalue computations. However, these algorithms assume fully error-corrected quantum hardware and are often difficult to implement in current NISQ devices due to limitations in circuit depth and quantum noise. In contrast to these gate-based approaches, quantum annealing (QA) is an alternative method for quantum optimization. QA leverages quantum tunneling to solve combinatorial optimization problems, which are formulated as quadratic unconstrained binary optimization (QUBO) or Ising models, and commercial annealing hardware has been developed primarily using D-Wave Systems. Several applications of QA for structural optimization involving discrete variables, such as member selection and topology optimization, have been reported. For example, Wils (2020) [31] demonstrated size optimization of two-dimensional truss structures using quantum annealing, and Honda et al. (2024) [32] proposed a QA-based method for truss structure optimization. However, QA has inherent limitations in terms of problem applicability, accuracy, and controllability.

To overcome these limitations, hybrid quantum algorithms that combine classical computing and quantum circuits, namely the class of variational quantum algorithms (VQAs), have been proposed. Peruzzo et al. (2014) [33] introduced the variational quantum eigensolver (VQE) for solving eigenvalue problems, and, in the same year, Farhi et al. (2014) [34] proposed the quantum approximate optimization algorithm (QAOA) for solving combinatorial optimization problems. McClean et al. (2016) [35] established the theoretical foundation of VQE, and Kandala et al. (2017) [36] demonstrated its feasibility on IBM’s NISQ hardware through experiments. McClean et al. (2018) [37] identified the barren plateau phenomenon, a critical limitation that hinders learning in deep circuits due to vanishing gradients. In response, Grimsley et al. (2019) [38] proposed the ADAPT-VQE algorithm, which incrementally constructs a circuit by adding necessary gates based on the problem rather than using a fixed ansatz. This approach improves the expressivity and trainability of the VQE. Building on these developments, practical applications of structural analysis have also been expanding. Liu et al. (2024) [39] experimentally implemented eigenfrequency estimation in structural analysis by combining ABAQUS with VQE in a hybrid quantum–classical pipeline. Sato et al. (2023) [40] proposed a VQE formulation based on the Rayleigh quotient variational principle, and Lee and Kanno (2023) [41] applied a QPE-based eigenvalue computation method to structural dynamics. In the 2020s, hardware-efficient adaptive ansatz variants, such as Qubit-ADAPT-VQE, have been proposed. Recently, Kim et al. (2024) [42] demonstrated a VQE implementation using high-dimensional degrees of freedom of a single-photon qudit. Other efforts have also contributed to improving the VQE performance, including optimization techniques, such as quantum natural gradient and SPSA, as well as measurement reduction strategies, such as Hamiltonian term grouping and quantum shadow methods. As the theoretical foundations and hardware feasibility for VQA continue to mature, efforts to integrate these techniques with deep learning have also become increasingly active.

Physics-Informed Neural Networks (PINNs), first proposed by Raissi et al. (2017, 2019) [43,44] and Karniadakis (2021) [45], combine numerical analysis with neural networks and have been effectively applied to partial differential equations (PDEs) in various spatial and temporal domains. Lu et al. (2021) [46] later systematized the implementation of PINNs through the DeepXDE framework, and Wang et al. (2021) [47] discussed their limitations and directions for improvement to support practical applications. In addition to the increasing adoption of PINN-based methods, research on Quantum PINNs (QPINNs), which integrate quantum circuits with neural networks, has also gained momentum. Markidis (2022) [48] proposed a continuous-variable quantum circuit (CVQC)-based QPINN for solving the one-dimensional Poisson equation and highlighted the unique loss landscape in quantum training by showing that stochastic gradient descent (SGD) outperforms Adam in terms of convergence stability. Trahan et al. (2024) [13] applied a variational quantum circuit (VQC) and compared pure QPINNs, pure PINNs, and hybrid PINNs, demonstrating that quantum nodes can achieve similar or better accuracy with fewer parameters than classical counterparts. Xiao et al. (2024) [49] introduced a PI-QNN architecture for PDEs with periodic solutions, achieving significantly lower prediction errors than conventional PINNs. Sedykh et al. (2023) [50] proposed a hybrid QPINN (HQPINN) for fluid dynamics problems with complex geometries, demonstrating the scalability of QPINNs to more complicated structural domains. In addition, Norambuena et al. (2024) [51] applied PINN techniques to quantum optimal control and achieved higher success rates and shorter solution times in state-transition problems for open quantum systems.

However, despite its various advantages, QPINN is often inefficient for practical structural analysis because of its slow computational speed and complex network architecture. In particular, hybrid QPINNs, which interconnect quantum circuits and neural network layers in an alternating sequence (quantum–neural–quantum–neural), tend to increase the hardware resource consumption and optimization burden. Repeated insertion of quantum layers may also lead to error accumulation owing to noise. Moreover, clear limitations exist in terms of scalability to high-dimensional structural problems. To address these limitations, this study adopted a quantum neural network (QNN)-based approach, which is simpler in structure and more computationally efficient than QPINNs. QNNs can approximate input–output relationships without explicitly calculating complex differential terms, offering advantages in terms of computational speed and training stability, which are critical in structural analysis and optimization. The physical relationships in linear and nonlinear systems, such as truss structures, can be effectively modeled from a function-approximation perspective. Furthermore, by integrating advanced circuit optimization techniques, such as the quantum natural gradient and hardware-efficient ansatz, the applicability of QNNs to real-world structural systems continues to expand.

## 3. Force Method

### 3.1. Force Method and Solutions in Truss Analysis

A truss structure is modeled as an assembly of one-dimensional elastic elements that carry only axial forces and are connected at the nodes to form the structure. For each member defined over a domain x∈[0,L], the displacement u(x) under an external axial force $P_{x}$ is governed by the differential equation(2)$$EAu_{,xx}+P_{x}=0\;,\;x\in \left[0,L\right],$$
which represents a boundary value problem (BVP) with essential and natural boundary conditions. However, because a truss is a discrete system in which multiple members are connected at joints, the boundary conditions of each member cannot be applied independently. Instead, the entire structure must satisfy equilibrium, compatibility, and flexibility conditions.

Among the various methods for obtaining such solutions, the finite element method (FEM) is the most widely used. The FEM leads to the global stiffness formulation of the structure, resulting in the following total stiffness equation:(3)Kd=p,
where K denotes the global stiffness matrix, d is the nodal displacement vector, and p represents the nodal force vector. Equation (Equation 3) is essentially an assembled system of equations that satisfies the boundary conditions of the individual members. Matrix K characterizes the deformation behavior and load–displacement relationship of the structure, and it corresponds to the Hessian matrix of a potential energy function, exhibiting symmetry and positive definiteness.

However, evaluating the internal forces requires first solving for nodal displacements from K and subsequently transforming the results into member coordinates. Therefore, it is difficult to directly interpret the structural behavior from K alone. In contrast, the force method provides a more direct and intuitive relationship between member forces and loads, as well as between nodal displacements and elongations, when compared with the displacement-based method.

### 3.2. Basic Equations and Mechanical Behavior

The fundamental equations of the force method are derived from the relationships of equilibrium, compatibility, and flexibility [6]. They are given as follows.

Consider a truss structure defined in a *d*-dimensional spatial domain consisting of *b* members and *j* nodes, where *c* degrees of freedom are constrained. The external force vector p and internal force vector t satisfy the following equilibrium equations:(4)At=p.

Here, A denotes an equilibrium matrix. Meanwhile, the compatibility relationship defines the connection between the member elongation vector e and nodal displacement vector d, and it is given as(5)Bd=e.

Here, B is the compatibility matrix, which is related to the equilibrium matrix through a transpose relationship. Because $B^{T}=A$, both matrices have the same rank *r*. This relationship is derived from the principle of virtual work and is expressed as $\delta e^{T}t=\delta d^{T}p$, indicating interdependence between the two matrices.

Finally, the flexibility relationship connects the equilibrium and compatibility relations by relating the internal forces to elongations as follows:(6)Ft=e.

Here, the flexibility matrix F is a diagonal matrix, where each diagonal element represents the flexibility of an individual member.

### 3.3. Solutions: Bar-Force t and Displacement d

The general solution t of the equilibrium Equation (Equation 4) can be expressed as the sum of a particular solution $t_{p}$ and a complementary homogeneous solution $t_{s}$, i.e.,(7)$$t=t_{p}+t_{s}=A^{+}p+S\alpha .$$

Here, $t_{p}$ is obtained using the pseudo-inverse $A^{+}$, and $t_{s}$ lies in the null space of A, which is spanned by the columns of matrix S. Vector α denotes a set of scalar coefficients.

Substituting t into the flexibility Relation (Equation 6), the member elongation vector becomes the following:(8)$$e=F\left(t_{p}+S\alpha \right).$$

As the elongation vector must satisfy compatibility, it is orthogonal to the left null space of B, and, given $B=A^{T}$, it follows that $S^{T}e=0$. Substituting e yields the following expression for α:(9)$$\alpha =-{\left(S^{T}FS\right)}^{-1}S^{T}Ft_{p}.$$

By substituting Equation (Equation 9) into Equation (Equation 7), the general solution for internal force vector t becomes(10)$$t=A^{+}p-W_{s}{\left(S^{T}FS\right)}^{-1}S^{T}FA^{+}p.$$

Similar to t, the general solution, d of the compatibility Equation (Equation 5) can also be expressed as the sum of a particular solution dp and a complementary homogeneous solution dm. However, in kinematically indeterminate systems, it obtains a solution to Equation (Equation 4). Nevertheless, because truss structures are generally kinematically determinate systems, the particular solution corresponding to the external force given by $d_{p}=B^{+}e$ serves as a valid solution to Equation (Equation 5). Instead of solving Equation (Equation 5) directly, it is more efficient to utilize the direct relationship between displacement d and external force p.

In Equation (Equation 6), the flexibility matrix F is a full-rank diagonal matrix, which is invertible. Hence, the inverse $F^{-1}$ always exists. By substituting $t=F^{-1}e$ into Equation (Equation 5) and then substituting the result into Equation (Equation 4), the following relationship is obtained:(11)$$AF^{-1}Bd=p.$$

Therefore, d can be expressed as(12)$$\begin{array}{cc} d & ={\left(AF^{-1}B\right)}^{-1}p\\ \; & =B^{+}FA^{+}p. \end{array}$$

Based on this result, it follows that the stiffness matrix in Equation (Equation 3) is given by $K=AF^{-1}B$, and a solution can be obtained if K is invertible. Similarly, the general solution for the compatibility Equation (Equation 5) can be written as the sum of a particular solution $d_{p}$ and the complementary homogeneous solution $d_{m}$. For kinematically indeterminate systems, it may be difficult to determine a general solution to Equation (Equation 4). However, most truss structures are kinematically determined, allowing for a particular solution corresponding to the external force to be expressed as $d_{p}=B^{+}e$.

Rather than solving Equation (Equation 5) directly, it is often more efficient to use a direct relationship between displacements d and external loads p. Based on the relationships derived above, the global stiffness matrix can be expressed as(13)$$K=AF^{-1}B.$$

If the matrix K is invertible, the system has a unique solution.

In conclusion, the vectors t and d given in Equations (Equation 7) and (Equation 12) are solutions to the equilibrium and compatibility equations, namely Equations (Equation 4) and (Equation 5) (or Equation (Equation 11)). Therefore, the approximated functions $Q_{\theta }$ for t and d described in Section 4 must satisfy the constraints imposed by Equations (Equation 4) and (Equation 11).

## 4. QNN Model

### 4.1. Overall QNN Architecture

This section describes the overall architecture of the quantum neural network (QNN)-based structural analysis process proposed in this study. The flowchart presented in Figure 1 summarizes the workflow of the surrogate modeling process using QNN. The proposed indexed QNN is based on the principles of the variational quantum algorithm (VQA) and consists of three main stages: quantum encoding of the input data, optimization of a parameterized quantum circuit (variational quantum circuit, VQC), and generation of outputs via quantum measurement. The following subsection defines and explains the structure of the VQC, a core component of the QNN, along with the basic quantum operations that constitute it, such as rotation and entanglement gates.

Variational quantum algorithms (VQAs) are hybrid computational approaches that combine quantum computing with classical optimization techniques, solving optimization problems by leveraging the unitary properties of quantum systems. This concept was first introduced through the variational quantum eigenvalue (VQE) proposed by Peruzzo et al. in 2014 [33], and it has since expanded into various applications, such as the quantum approximate optimization algorithm (QAOA) and variational quantum classifier (VQC). Among these VQAs, the quantum neural network (QNN) represents a typical quantum–classical hybrid learning framework that operates based on fundamental quantum properties, such as superposition and entanglement. A QNN models the relationship between the input and output by constructing a quantum circuit in a neural network-like structure with a variational quantum circuit (VQC) composed of rotation gates ($R_{X}$, $R_{Y}$, $R_{Z}$) and entangling gates (e.g., CNOT) at its core. In this section, we introduce the definitions and mathematical formulations of the fundamental operations used in the QNN implementation: rotation and entangling gates. The rotation gate is a single-qubit unitary operation that rotates the qubit state about a specific axis, and it is defined as follows:(14)$$R_{\alpha }^{\left(i\right)}\left(\theta \right)=\exp\left(-i\frac{\theta }{2}\sigma_{\alpha }^{\left(i\right)}\right),\;\alpha \in \left\{X,Y,Z\right\}.$$

Here, *I* denotes the identity operator, and *X*, *Y*, and *Z* refer to the Pauli matrices. The $R_{Z}\left(\theta \right)$ gate applies only a phase shift to basis states |0〉 and |1〉. By contrast, the $R_{X}\left(\theta \right)$ and $R_{Y}\left(\theta \right)$ gates perform rotations in the |±〉 and |±i〉 eigenbases, respectively, thereby controlling the qubit state. The controlled-NOT (CNOT) gate is a two-qubit entangling operator that applies the Pauli-X operation (*X*) to the target qubit when the control qubit is in the |1〉 state, thereby inducing quantum correlations between the qubits. The *X* operation functions as a classical NOT gate by flipping the qubit state between |0〉 and |1〉. A CNOT gate is mathematically defined as follows [52]:(15)$$\mathrm{CNOT}=\left(\begin{array}{cccc} 1 & 0 & 0 & 0\\ 0 & 1 & 0 & 0\\ 0 & 0 & 0 & 1\\ 0 & 0 & 1 & 0 \end{array}\right)=e^{-i\frac{\pi }{4}\left(I_{1}-Z_{1}\right)\left(I_{2}-X_{2}\right)}.$$

Based on these fundamental quantum operations, this study designed two quantum neural network architectures to process indexed input data.

The quantum neural network (QNN) implemented in this study can be categorized into two architectures based on how the input data are processed. First, the single-domain structure encodes the entire input dataset into a unified variational quantum circuit (VQC) for processing. All input variables are mapped to a single quantum state, and the transformation and optimization of the entire data are carried out through a single VQC. This approach offers a relatively simple circuit design, and it is effective when the input dimension is low or the structural complexity is minimal Figure 2a.

In contrast, the separate-domain structure divides the input data by index domain and constructs independent quantum circuits for each domain, allowing them to be trained separately. Each circuit performs localized optimization on a specific subset of the input data, and the final output is composed by aggregating the measurement results from each circuit. This approach is particularly suitable for handling high-dimensional input data or large-scale structural analysis problems (Figure 2b).

As illustrated in Figure 2, the quantum neural network (QNN) consists of three stages: (1) a feature map (input encoding), which transforms classical inputs into quantum states; (2) variational quantum circuit (parameterized circuit), denoted as $U_{\theta }$, which includes parameterized rotation gates and entangling gates such as CNOT; and (3) measurement (output decoding), which extracts classical outputs from quantum states via expectation value measurements.

In particular, the entanglement between qubits in the VQC is implemented through the following sequence of operations:(16)$$\varepsilon^{\left(l\right)}=\mathrm{CNOT}\left(1,2\right)\cdot \mathrm{CNOT}\left(2,3\right)\cdots \mathrm{CNOT}\left(n-1,n\right).$$

The entangling operation $\varepsilon^{\left(l\right)}$ is composed of a sequence of CNOT gates applied between adjacent qubit pairs, which effectively generates quantum entanglement within the circuit. This entanglement structure enhances the expressivity of the quantum circuit by enabling interactions between qubits, allowing the model to effectively capture complex data characteristics. The following section provides a detailed explanation of the input encoding method, the parameterized structure of the variational quantum circuit, and the measurement and output process, which together constitute the QNN architecture.

### 4.2. Input Representation: Index-Based Encoding

In this study, the proposed quantum neural network (QNN) encodes input data in an index-based structure and constructs a parameterized quantum circuit (variational quantum circuit, VQC) accordingly.

A representative encoding technique involves mapping each input value $x_{i}$ onto a qubit using the $R_{Y}$ rotation gate. The full input vector $x={\left[x_{1},x_{2},\cdots ,x_{n}\right]}^{T}$ is thus transformed into a quantum state as follows:(17)$$|\psi \rangle =\overset{n}{\underset{i=1}{⨂}}R_{Y}^{\left(i\right)}\left(x_{i}\right)|0\rangle .$$

Here, $R_{y}^{\left(i\right)}\left(x_{i}\right)$ denotes the operator that applies a single-qubit rotation around the *Y*-axis to the *i*-th qubit based on the input value $x_{i}$, transforming the initial state ${|0\rangle }^{⊗n}$ into a quantum state that reflects the input.

The external load vector $p={\left\{p_{j}\in R^{n^{d}}\right\}}_{j\in N}$ returns the force vector $p_{j}$ applied at each node according to the given conditions. Each member i∈M returns its connectivity information via the mapping $C\left(i\right)=\left(a(i),\;b(i)\right)$, where the connectivity structure is defined as $C:M\to N\times N,\;N\times N=\left\{(i,j)|i\in N,\;j\in N\right\}$. The axial rigidity of each member is represented by the index-based expression ${\mathrm{EA}}^{\left(i\right)}={\left\{EA^{\left(i\right)}\right\}}_{i\in M}$.

The feature vector ϕ(i) constructed from these index-based inputs is used to define the loss function of the quantum neural network model. Depending on the prediction target, the composition of ϕ(i) varies. For example, the model predicting axial force $\hat{t}\left(i\right)$ is defined as follows:(18)$$t_{i}=Q_{\theta }^{t}\left(\phi_{t}\left(i\right)\right),\;\mathrm{where}\;\phi_{t}\left(i\right)=\left(x_{a(i)},\;x_{b(i)},\;p_{a(i)},\;p_{b(i)},\;EA_{i}\right).$$

The model for predicting nodal displacements $\hat{d}\left(j\right)$ is defined as follows:(19)$$d_{j}=Q_{\theta }^{d}\left(\phi_{d}\left(j\right)\right),\;\mathrm{where}\;\phi_{d}\left(j\right)=\left(x_{j},\;p_{j},\;\mathrm{EA}\right).$$

Here, both quantum neural network models $Q_{\theta }^{t}$ and $Q_{\theta }^{d}$ are defined as $\hat{Q}:N^{+}\to R$, where the parameter space is replaced with index-based inputs, effectively reducing the input dimensionality. Compared to coordinate-based input models, such as $Q_{\theta }:R^{2n^{d}}\to R$ or $Q_{\theta }:R^{n^{d}}\to R^{n^{d}}$, this formulation is significantly more concise and enables improved scalability and flexibility of the output structure.

In addition, when using the member set M or node set N as a dataset for a single epoch, the quantum neural network can be trained without explicitly constructing separate mechanical information for each index since the equilibrium matrix A and compatibility matrix B can be directly utilized. This allows for a more efficient modeling process within the quantum neural network framework.

### 4.3. VQC (Variational Quantum Circuit)

This section explains how a variational quantum circuit (VQC) is constructed for the encoded quantum state. The core component of a quantum neural network (QNN), the variational quantum circuit (VQC), is constructed by repeatedly applying trainable rotations and entangling gates. The overall circuit is expressed as follows:(20)$$U\left(\theta \right)=\prod_{l=1}^{L}\left(\overset{n}{\underset{i=1}{⨂}}R_{Z}^{\left(i\right)}\left(\theta_{l,i}^{\left(Z\right)}\right)R_{Y}^{\left(i\right)}\left(\theta_{l,i}^{\left(Y\right)}\right)R_{X}^{\left(i\right)}\left(\theta_{l,i}^{\left(X\right)}\right)\right)\cdot \varepsilon^{\left(l\right)}.$$

Here, *L* denotes the number of layers, *n* represents the number of qubits, and $\varepsilon^{\left(l\right)}$ refers to the entangling operation in the *l*th layer, which is composed of a sequence of CNOT gates between adjacent qubits. The rotation gate block $R_{Z}^{\left(i\right)}R_{Y}^{\left(i\right)}R_{x}Z\left(i\right)$ is a sequence of single-qubit unitary operations applied to the *i*th qubit in the *l*th layer, rotating the qubit state around the *Z*, *Y*, and *X* axes. This combination forms a single SU(2) unitary operator, allowing flexible control over the position of the qubit on the complex spherical coordinate system.(21)$$R_{\alpha }^{\left(i\right)}\left(\theta \right)=\exp\left(-i\frac{\theta }{2}\sigma_{\alpha }\right).$$

Therefore, applying this sequence of three rotation operations to each qubit increases the expressivity of the quantum circuit. For instance, in the case of a circuit composed of two qubits (n=2) and two layers (L=2), the parameterized circuit U(θ) is expanded as follows:(22)$$U\left(\theta \right)=\left(\overset{2}{\underset{i=1}{⨂}}R_{z}^{\left(i\right)}\left(\theta_{2,i}^{\left(z\right)}\right)R_{y}^{\left(i\right)}\left(\theta_{2,i}^{\left(y\right)}\right)R_{x}^{\left(i\right)}\left(\theta_{2,i}^{\left(x\right)}\right)\right)\cdot \mathrm{CNOT}\left(1,2\right)\cdot \left(\overset{2}{\underset{i=1}{⨂}}R_{z}^{\left(i\right)}\left(\theta_{1,i}^{\left(z\right)}\right)R_{y}^{\left(i\right)}\left(\theta_{1,i}^{\left(y\right)}\right)R_{x}^{\left(i\right)}\left(\theta_{1,i}^{\left(x\right)}\right)\right)\cdot \mathrm{CNOT}\left(1,2\right).$$

Each layer consists of a sequence of parameterized single-qubit rotation gates ($R_{x}$, $R_{y}$, $R_{z}$), which is followed by an entangling gate (CNOT). The independent parameters $\theta_{l,i}^{\left(k\right)}$ are used for each qubit *i* and rotation axis *k* in the layers l=1 and 2. For a two-qubit system, the entangled structure is composed of a single CNOT(1,2) gate, which is applied identically across all the layers. Finally, applying the full quantum circuit to the encoded input state |x〉 yields the following trainable quantum state:(23)|ψ(θ)〉=U(θ)|x〉.

### 4.4. Measurement and Output

The quantum state that passes through the VQC is measured with respect to a specific observable *P*, and its expectation value is used as the output. The entire circuit generates a quantum state |ψ(θ,x)〉 depending on the parameter vector $\theta =\theta_{l,i}^{\left(x\right)},\theta_{l,i}^{\left(y\right)},\theta_{l,i}^{\left(z\right)}$ and the input x. The expectation value was computed as follows:(24)〈P〉=〈ψ(θ,x)|P|ψ(θ,x)〉.

The measurement operator *P* is typically given in the form of a tensor product of Pauli operators applied to selected qubits, such as $P=Z^{\left(i\right)}$ or $P=Z^{⊗n}$. The resulting expectation value 〈P〉 is interpreted as the classical output $y_{\mathrm{pred}}$ and used to compute the loss function L by evaluating the discrepancy from the target value $y_{\mathrm{true}}$ as follows:(25)$$L_{r}\left(\theta \right)=\frac{1}{N}\sum_{j=1}^{N}{\left({\langle P\rangle }_{j}-y_{j}^{\mathrm{true}}\right)}^{2}.$$

This loss function is iteratively minimized using a classical optimizer that updates the values of the trainable parameters θ. This process forms the core structure of the variational quantum algorithm (VQA), consisting of the iterative cycle of “quantum state preparation → measurement → loss evaluation → parameter update”. By optimizing the multivariable parameters θ, the quantum circuit learns a suitable representation of the target problem.

### 4.5. Truss Analysis Model and Loss Function

#### 4.5.1. Mathematical Model for Truss Structural Analysis

To design the loss function within the QNN framework described in the previous section, the static truss analysis model introduced in Section 3 is formulated as follows:(26)$$\mathrm{Find}\;u\in R^{n}\;\mathrm{such}\;\mathrm{that}\;A\left(u\left(X,z\right)\right)=0.$$

Here, A denotes a mechanically defined operator, $X\in \Omega_{s}$ represents the spatial domain, and $z\in \Omega_{p}$ denotes the set of physical parameters of the truss structure. In Equation (Equation 26), the member force model corresponds to u←t, which satisfies the equilibrium Equation (Equation 4), while the nodal displacement approximation model is defined as u←d, satisfying the stiffness Equation (Equation 11).

The quantum neural network (QNN) model $Q_{\theta }$ used to approximate such boundary value problems in a finite-dimensional setting can be expressed as follows:(27)$$u\left(X,z\right)\approx \hat{u}\left(X,z,\theta \right)=Q_{\theta }\left(X,z\right)\;,\;X\in \Omega_{s}\;,\;z\in \Omega_{p}.$$

Here, $\Omega_{s}=\left\{X\in R^{dj-c}|X_{j}\in R^{d},\;j=1,\ldots ,m\right\}$ represents the spatial domain, $\Omega_{p}\subseteq R^{b\times p}$ denotes the set of physical parameters for the truss structure, and θ refers to the parameters of the QNN.

In the separate-domain structure, the input consists of index-based data, as in the single-domain case, but it is partitioned according to the corresponding indices for each separated QNN structure and embedded into each quantum circuit individually. In this case, as illustrated in Figure 1, the loss function is computed by aggregating the outputs from each quantum circuit, which means that the output dataset must preserve the mechanical information throughout. Therefore, training is conducted using the output t corresponding to the index dataset M for a single epoch (or, alternatively, using N and d).

#### 4.5.2. Largrangian Dual Optimization

To design a loss function for the QNN model that predicts either the member forces or nodal displacements, the problem can be formulated as an optimization problem consisting of a residual term, which is expressed using the Euclidean norm and mechanical equality constraints.(28)$$\begin{array}{cc} \min & \;{∥u-\hat{u}∥}_{2}^{2}\\ \mathrm{subject}\;\mathrm{to} & \;A\hat{u}=p, \end{array}$$
where $\hat{u}$ denotes the intended solution obtained using Equation (Equation 27), A is the coefficient matrix of the equilibrium equation (or the coefficient matrix from Equation (Equation 11)), and p represents the external force vector. One possible approach for solving the equality-constrained optimization problem defined in Equation (Equation 28) is to reformulate it as an unconstrained optimization problem using a quadratic penalty function.

Let the objective function be defined as $J_{r}\left(\theta \right)={∥u-\hat{u}∥}_{2}^{2}$ and the mechanical constraint function as(29)$$C\left(\theta \right)=p-A\hat{u}.$$

Then, using $J_{r}$, *C* and the penalty parameter β, Equation (Equation 28) can be reformulated as the following unconstrained optimization problem:(30)$$\arg\min_{\theta }J_{\beta }\left(\theta \right)=\arg\min_{\theta }J_{r}\left(\theta \right)+\beta {∥C(\theta )∥}_{2}^{2},$$
where β is a penalty parameter that follows an increasing scalar sequence across the iterations. Although the value of β significantly influences the stability of the optimization system, stability analysis is beyond the scope of this study.

The optimization model in Equation (Equation 30) represents an approach for solving the constrained optimization problem. An alternative method for solving Equation (Equation 28) is the Lagrangian multiplier technique, which is formulated as follows:(31)$$\arg\min_{\theta }J_{\lambda }\left(\theta \right)=\max_{\lambda \geq 0}\min_{\theta }J_{r}\left(\theta \right)+\lambda \cdot C\left(\theta \right).$$

Here, $\lambda \in R^{N}$ is the Lagrangian multiplier vector, which is updated at each iteration according to the learning rate η as follows:(32)$$\lambda^{\left(t+1\right)}=\lambda^{\left(t\right)}+\eta \;C\left(\theta^{\left(t\right)}\right).$$

This method provides a method to obtain an optimal solution by formulating the original optimization problem together with its dual problem.

The augmented Lagrangian method, which combines the two aforementioned approaches, is described in [53]. This method augments the Lagrangian function $J_{\lambda }\left(\theta \right)$ by incorporating an additional penalty term, and it is formulated as follows:(33)$$\arg\min_{\theta }J_{\lambda ,\beta }\left(\theta \right)=\max_{\lambda \geq 0}\min_{\theta }J_{r}\left(\theta \right)+\lambda \cdot C\left(\theta \right)+\beta {∥C(\theta )∥}_{2}^{2}.$$

The sequence of updating λ to obtain the optimal solution in Equation (Equation 33) is defined as(34)$$\lambda^{\left(t+1\right)}=\lambda^{\left(t\right)}+\beta \;C\left(\theta^{\left(t\right)}\right).$$

This type of approach is known to converge stably, without requiring an increasing sequence of β values [53].

In this study, a loss-function-based optimization model was constructed for the structural analysis of truss systems. The variational quantum algorithm (VQA) encodes the input data into quantum states through a variational quantum circuit (VQC) and computes the loss function from the measured output. The parameters are then iteratively optimized using a classical optimizer, specifically the Adam algorithm.

#### 4.5.3. Loss Functions for the Prediction of Bar Force

Based on the Lagrangian dual-optimization framework, the loss function for the member force prediction model was defined. First, the residual loss function is given by(35)$$L_{r}\left(\theta \right)=\frac{1}{N}{∥t-\hat{t}∥}_{2}^{2}=\frac{1}{N}{∥r∥}_{2}^{2},$$
where $r=t-\hat{t}$ and $\hat{t}$ denote the predicted values obtained through using $Q_{\theta }$. The quadratic penalty term is defined as follows:(36)$$L_{q}\left(\beta ,\theta \right)=\beta {∥p-A\hat{t}∥}_{2}^{2},$$
where β is the penalty parameter. The loss term $L_{q}$ quantifies the degree of violation of the structural equilibrium condition, and it serves as an additional constraint to ensure structural consistency. Given that p=At, Equation (Equation 26) can be equivalently expressed as(37)$$L_{q}\left(\beta ,\theta \right)=\beta {∥A(t-\hat{t})∥}_{2}^{2}=\beta {∥Ar∥}_{2}^{2}.$$

The loss function term that incorporates the Lagrangian multiplier is defined as follows:(38)$$L_{L}\left(\lambda ,\theta \right)=\lambda^{T}\left(p-A\hat{t}\right)=\lambda^{T}\left(Ar\right),$$
where $\lambda^{T}$ is the Lagrangian multiplier vector, and Ar represents the constraint term that directly enforces the equilibrium condition through the Lagrangian formulation.

The total loss function $L_{r}$, $L_{q}$, and $L_{L}$, as well as the total loss function $L_{t}$, can be formulated, where the optimization model based on the quadratic penalty method is expressed, as follows:(39)$$L_{t}^{\beta }\left(\beta ,\theta \right)=L_{r}+L_{q}=\frac{1}{N}{∥r∥}_{2}^{2}+\beta {∥Ar∥}_{2}^{2},$$
where penalty parameter β is treated as a constant. The optimization model that incorporates the Lagrangian multiplier is formulated as follows:(40)$$L_{t}^{\lambda }\left(\lambda ,\theta \right)=L_{r}+L_{L}=\frac{1}{N}{∥r∥}_{2}^{2}+\lambda^{T}\left(Ar\right).$$

The Lagrangian multiplier λ is updated as follows:(41)$$\lambda_{i+1}=\lambda_{i}+\eta \left(Ar\right).$$

The augmented Lagrangian method, which combines $L_{t}^{\beta }$ and $L_{t}^{\lambda }$, is defined as follows:(42)$$\begin{array}{cc} L_{t}^{\lambda ,\beta }\left(\lambda ,\beta ,\theta \right) & =L_{r}+L_{L}+L_{q}\\ \; & =\frac{1}{N}{∥r∥}_{2}^{2}+\lambda^{T}\left(Ar\right)+\beta {∥Ar∥}_{2}^{2}. \end{array}$$

The Lagrangian multiplier λ used in $L_{t}^{\lambda ,\beta }$ is updated as follows:(43)$$\lambda_{i+1}=\lambda_{i}+\beta \left(Ar\right).$$

The model proposed in this study applies the same optimization framework to both the member force t and displacement d QNN models, except for the design variables, as follows. The residual for d is given by $r=d-\hat{d}$, and the coefficient matrix becomes $A\leftarrow [AF^{-1}B]=K$. Accordingly, the residual loss is $L_{r}=\frac{1}{N}{\left|r\right|}_{2}^{2}$ and the total loss function is defined as(44)$$\begin{array}{cc} L_{t}^{\beta }\left(\beta ,\theta \right) & =L_{r}+\beta {∥Kr∥}_{2}^{2}\\ L_{t}^{\lambda }\left(\lambda ,\theta \right) & =L_{r}+\lambda^{T}\left(Kr\right)\\ L_{t}^{\lambda ,\beta }\left(\lambda ,\beta ,\theta \right) & =L_{r}+\lambda^{T}\left(Kr\right)+\beta {∥Kr∥}_{2}^{2}. \end{array}$$

The updates for the Lagrangian multiplier and penalty parameter in each loss function are the same as those used in the member force model.

## 5. Numerical Examples

All of the experiments in this study were conducted on a macOS system equipped with a 14-core CPU, 20-core GPU, and a 16-core Neural Engine (Table 1). The implementation was based on Python 3.10, and the main software libraries used were NumPy (1.26.4) and PennyLane (0.40.0). All quantum circuit construction and optimization were performed using PennyLane’s default.qubit simulator.

The performance of each loss function case was evaluated using the mean squared error ($L_{2}$ error) and the coefficient of determination ($R^{2}$). The $R^{2}$ score and the $L_{2}$ error are defined as follows:(45)$$R^{2}=1-\frac{\sum {\left(y_{\mathrm{target}}-y_{\mathrm{pred}}\right)}^{2}}{\sum {\left(y_{\mathrm{target}}-\bar{y}\right)}^{2}},$$(46)$$L_{2}\;\mathrm{error}=\frac{1}{N}\sum_{i=1}^{N}{\left(y_{\mathrm{target}}-y_{\mathrm{pred}}\right)}^{2}.$$

Here, $y_{\mathrm{pred}}$ denotes the predicted value, $y_{\mathrm{target}}$ refers to the target value, *N* indicates the total number of samples, and $\bar{y}$ represents the mean value of $y_{\mathrm{target}}$. A summary of the truss and dome models used in this study is presented in Table 2.

This section validates the proposed index-based single and separate-domain QNN architecture by applying it to various numerical examples and analyzing prediction accuracy and loss functions. In particular, the number of parameters is compared according to the number of qubits and layers, and the model performance is quantitatively evaluated in terms of convergence characteristics and accuracy based on the $L_{2}$ error.

### 5.1. 10-Bar Plane Truss

This section presents the structural analysis of the 10-bar plane truss. As shown in Figure 3, the model consists of 10 linear members and 6 nodes, with both horizontal and vertical distances between nodes set to L=1m. Vertical loads of $p_{3y}=-1\;\mathrm{kN}$ and $p_{5y}=-1\;\mathrm{kN}$ are applied to Node 3 and Node 5, respectively. The objective of this analysis is to predict the axial forces in all 10 members. After encoding the input data for each member, the axial forces are predicted using a single-domain QNN model, as illustrated in Figure 2a. The material properties used in the truss model are as follows: Young’s modulus $E=2.0\times 10^{11}\;{N/m}^{2}$, cross-sectional area $A=1.0\;{\mathrm{cm}}^{2}$, and density $\rho =7860\;{\mathrm{kg}/m}^{3}$. The training was conducted using the Adam optimizer with a learning rate of 0.05 for up to 5000 epochs.

The 10-bar plane truss, as a simple two-dimensional structure, was used to focus on analyzing the performance with respect to the number of qubits and layers. To this end, Case 1 was evaluated using various trainable single-domain quantum circuit configurations combining 2–11 qubits and 2–11 layers.

The analysis results for Case 1 are summarized in Table 3 for the range of 2–6 qubits and 2–6 layers. As the number of qubits and layers increased, the $L_{2}$ error decreased, showing a tendency of improved prediction performance. The coefficient of determination $R^{2}$ increased with deeper layers, and most configurations with six or more layers reached $R^{2}\approx 1.0$. In particular, both the 5Q–6L and 4Q–6L configurations achieved $R^{2}=1.0$. As shown in Figure 4, the results in Area 2 exhibited satisfactory performance with $R^{2}$ values approaching 1.0, and, within the Area 1 region, a tendency of improved prediction performance was also observed as the number of qubits and layers increased. The time values in Table 3 represent the time required to complete the entire training for each qubit–layer configuration and are consistently used throughout this paper. However, simply increasing the number of qubits and layers did not always lead to improved performance, and, as shown in Figure 4a, configurations beyond 6Q–6L exhibited either reduced prediction performance or lower training efficiency. In particular, in the 7–11 qubit range, a tendency of decreasing prediction performance was observed as the number of layers increased.

The convergence curve illustrates how the $L_{2}$ error decreases with training epochs and is used to assess the model’s training stability and accuracy. Figure 5a–e shows the convergence curves for each qubit–layer configuration in Case 1. Figure 5f visualizes the loss function behavior of the 6-layer models, which showed the best performance for each qubit count. To evaluate the effectiveness of the proposed quantum neural network (QNN) architecture, a comparative analysis was also conducted using a classical neural network (NN) model on the same 10-bar plane truss problem.

The performance of a classical neural network (NN) model corresponding to Table 3 is presented in Figure 6. To enable a fair comparison, the number of layers and the number of neurons per layer in the fully-connected NN were set equal to the number of layers and qubits in the QNN. The training settings, including the Adam optimizer and learning rate, were also matched, with the activation function set to tanh. Figure 6a shows the $L_{2}$ error of the NN model ($L_{2}^{\left(D\right)}$), while (b) illustrates the difference between the NN and QNN errors ($L_{2}^{\left(D\right)}-L_{2}^{\left(Q\right)}$). A positive value indicates that the QNN achieved superior prediction accuracy. Figure 6c depicts the parameter ratio $n^{Q}/n^{D}$, where values less than 1.0 indicate that the QNN used fewer parameters than the NN.

- 0pt

According to Table 3, the QNN achieved the lowest $L_{2}$ error when the number of layers was six, and the best performance was particularly exhibited with four qubits. Compared to the NN with a 6-layer, 4-neuron configuration, the QNN achieved better prediction results with only about 64% of the total parameters. Even for the 6-layer, 5- and 6-neuron NN configurations, which showed relatively good performance, the QNN achieved lower $L_{2}$ errors (e.g., $4.79\times 10^{-4}$) with only about 50% of the parameters.

Figure 7 visualizes the distribution of the coefficient of determination $R^{2}$, $L_{2}$ error, and the epoch at which the minimum $L_{2}$ error was reached for the 2–6 qubit and 2–6 layer combinations in Case 2–4. This confirms that, as the number of qubits and layers increased, both the overall prediction performance and convergence speed improved.

As shown in Table 3, most models in Case 1 achieved high prediction accuracy based on $R^{2}$ within the 2–6 qubit and 2–6 layer range. In particular, all of the configurations converged to at least $R^{2}\geq 9.990\times 10^{-1}$, with some reaching $R^{2}=1.00$.

For Cases 2–4 also, as presented in Figure 7, all of the models within the Area 1 configuration (2–6 qubits and 2–6 layers) recorded excellent prediction accuracy with $R^{2}>0.998$. Figure 7 visualizes the $R^{2}$ convergence curves, $L_{2}$ error values, and the minimum $L_{2}$ error epoch across different qubit–layer configurations for each case. Figure 8 provides a comparison of the predicted values $y_{\mathrm{pred}}$ and the target values $y_{\mathrm{target}}$.

Therefore, the Area 1 range (2–6 qubits and 2–6 layers) was adopted as the design space for the analysis of Cases 2–4, including the common configuration of Area 3 (6 qubits and 6 layers). The detailed model configurations and experimental scopes for each single-domain problem are explained in the subsequent sections.

### 5.2. 25-Bar Plane/Space Truss

#### 5.2.1. 25-Bar Plane Truss

The structural analysis of the 25-bar planar truss presented in Figure 9 is performed and discussed in this section. The target model consists of a total of 25 members and 14 nodes, where seven nodes are uniformly arranged in both the horizontal and vertical directions on the top and bottom with a spacing of L=9.144m. Accordingly, the total length and height of the structure are 54.864m and 9.144m, respectively. Vertical loads are applied in the *y*-direction at the top nodes of the truss, with $P_{1y}=P_{7y}=-400.2\;\mathrm{kN}$ applied at Nodes 1 and 7, and $P_{2y∼6y}=-800.4\;\mathrm{kN}$ applied at Nodes 2 through 6. All members share identical material properties: Young’s modulus $E=69\times 10^{9}\;{N/m}^{2}$, cross-sectional area $A=0.01\;m^{2}$, and density $\rho =2700\;{\mathrm{kg}/m}^{3}$. The objective of this example was to predict the axial force developed in each member under the given loading conditions. The analysis was conducted using the single-domain model shown in Figure 2a and the separate-domain model shown in Figure 2b. The models were trained using the Adam optimizer with a learning rate of 0.05 for 10,000 iterations. For the single-domain condition, Case 1 was analyzed across a range of 2–6 qubits and 2–20 layers to enable parameter comparisons.

Table 4 presents the $R^{2}$ scores corresponding to various 2–6 qubit combinations under both the single-domain and split-domain settings. Notably, when the structure was divided into five domains, the 2-qubit–3-layer (2Q–3L) configuration achieved excellent performance with an $L_{2}$ error of $2.8102\times 10^{-14}$ and $R^{2}=1.000$ in only 526 epochs. In contrast, the 3-qubit–14-layer (3Q–14L) configuration under the single-domain setting required 6193 epochs to reach a comparable accuracy level with an $L_{2}$ error of $4.2138\times 10^{-11}$. Furthermore, the split-domain model used only 90 parameters, which is approximately 6.25% fewer than the 96 parameters required in the single-domain case.

Based on these results, the same experimental settings applied to the 10-bar planar truss were extended to the 25-bar planar and space truss models. For Cases 2–4, a restricted hyperparameter range of 2–6 qubits and 2–6 layers was used, with training conducted under the split-domain strategy. Figure 10 compares the performance between the single-domain and multi-domain settings for Case 1.

According to the results in Table 5 for Case 2, the single-domain 2-qubit–16-layer (2Q–16L) model required a total of 96 parameters and 822 epochs to reach an $L_{2}$ error of $2.4883\times 10^{-12}$. In contrast, the 2-qubit–3-layer (2Q–3L) model trained on five split domains achieved a lower $L_{2}$ error of $3.4959\times 10^{-14}$ in just 533 epochs using only 90 parameters—showing superior convergence despite a 6.25% reduction in parameter count.

A similar trend was observed in the Case3 results, as shown in Table 6. Under the condition $\beta =1\times 10^{-6}$, the single-domain 3-qubit–18-layer (3Q–18L) model required 162 parameters and 6027 epochs to reach an $L_{2}$ error of $3.6791\times 10^{-8}$. In contrast, the split-domain 2-qubit–4-layer (2Q–4L) model used only 120 parameters to achieve a significantly lower $L_{2}$ error of $2.6853\times 10^{-9}$ in just 2615 epochs.

The results in Table 7 also confirm the learning efficiency of the separate-domain approach in Case 4. In the single-domain setting, the 3-qubit–19-layer (3Q–19L) model required 2421 epochs and 171 parameters to reach an $L_{2}$ error of $2.9086\times 10^{-8}$ and $R^{2}=1.0$. Meanwhile, the 2-qubit–5-layer (2Q–5L) model trained with five domains reached a lower $L_{2}$ error of $1.8233\times 10^{-9}$ and $R^{2}=1.0$ in just 183 epochs using only 150 parameters. These results demonstrate that, even with a reduced number of parameters, the domain-splitting strategy yields significantly improved training speed and prediction accuracy. On the other hand, under the condition β=0.01, the $L_{2}$ error converged to a relatively large value, indicating unsatisfactory performance.

Figure 11 and Figure 12 visualize the comparison between the predicted and true values and the $L_{2}$ error convergence behavior under the condition of training with five subdomains at $\beta =1\times 10^{-6}$. Among the tested configurations, the 2Q–3L model in Case 2 converged to $L_{2}=3.4959\times 10^{-14}$ at 533 epochs, and the 2Q–4L model in Case 3 reached $L_{2}=2.6853\times 10^{-9}$ at 2615 epochs. Meanwhile, the 2Q–5L configuration in Case 4 achieved $L_{2}=1.8233\times 10^{-9}$ in just 183 epochs.

#### 5.2.2. 25-Bar Space Truss

This section presents the analysis of a 25-bar space truss structure and discusses the results. The target structure consists of 10 nodes and 25 members distributed in the 3D *x*–*y*–*z* space. Nodes 1 and 2 are located at the top with a height of H=2.5m, while fixed support conditions are applied to the four bottom nodes (Nodes 7–10). The remaining nodes (Nodes 3–6) are placed at the intermediate height z=0, forming an overall symmetric structure. The nodal coordinates are defined based on a horizontal spacing of $L_{1}=1.0\;m$, a diagonal length $L_{2}=2.5\;m$, and a vertical height H=2.5m. External loads are applied to the top Nodes 1 and 2, with magnitudes of $p_{1y}=80\;\mathrm{kN}$, $p_{1z}=-20\;\mathrm{kN}$, $p_{2y}=-80\;\mathrm{kN}$, and $p_{2z}=-20\;\mathrm{kN}$. The material properties applied to the structure are as follows: Young’s modulus $E=2.0\times 10^{11}\;{N/m}^{2}$, density $\rho =7860\;{\mathrm{kg}/m}^{3}$, and cross-sectional area $A=1.0\times 10^{-5}\;m^{2}$. The objective of this example was to predict the axial force developed in each member under the given loading conditions. Training was performed using the Adam optimizer with a learning rate of 0.05 for up to 10,000 iterations.

Based on the analysis results of the 25-bar truss problem, the 25-member space truss was analyzed by dividing it into five domains (Figure 13). As shown in Table 8, both Case 1 and Case 2 employed the 2Q–3L configuration with a total of 90 parameters. The $L_{2}$ errors converged to $3.4760\times 10^{-13}$ after 1208 training epochs for Case 1 and to $2.9885\times 10^{-13}$ after between 1236 and 1975 training epochs for Case 2. In Cases 3 and 4, although the number of parameters increased to the range of 180–375, satisfactory results were still obtained. Notably, in Case 4, the 3Q–5L configuration with 225 parameters achieved convergence to an $L_{2}$ error of $3.9648\times 10^{-10}$ over 9478 epochs.

Figure 14 illustrates the $L_{2}$ error convergence for the 25-bar space truss under the condition of $\beta =1\times 10^{-6}$, while Figure 15 presents the comparison between the predicted values $y_{\mathrm{pred}}$ and the exact results $y_{\mathrm{exact}}$ for each case. In Case 3, the 2Q–6L configuration, with a total of 180 parameters, achieved an $L_{2}$ error of $7.4887\times 10^{-10}$ within 1630 epochs. For Case 4, the 3Q–5L model reached an $L_{2}$ error of $3.9648\times 10^{-10}$ at 9478 epochs. In Cases 1 and 2 converged to an $L_{2}$ error on the order of $10^{-13}$ within 1208 and 1236 epochs, respectively, yielding satisfactory results.

### 5.3. 6-by-6 Square Grid Dome

Figure 16 consists of a 6×6 grid of nodes arranged at uniform intervals of L=5.0m. The total number of nodes in the figure is 85, and the structure is composed of 288 members. The curved surface geometry was generated in the form of z=f(x,y), with the maximum deflection occurring at the center of the structure. This study addresses the problem of predicting axial forces under varying loading conditions and geometries, and it compares the performance of QNN models under different loss function settings. Vertical loads in the *z*-direction were applied to the nodes, with fixed boundary conditions imposed on the bottom nodes. A uniform external load of −30kN was applied either at the top or bottom nodes. All members in this model share identical material properties, with a Young’s modulus $E=2.0\times 10^{11}\;{N/m}^{2}$, a density $\rho =7860\;{\mathrm{kg}/m}^{3}$, and a cross-sectional area $A=1.0\times 10^{-3}m^{2}$. Model training was conducted using the Adam optimizer with a learning rate of 0.05 for up to 10,000 iterations.

The axial force analysis results for the 6-by-6 grid dome structure are summarized in Table 9, and the prediction results are shown in Figure 17. In Case 1, the 5Q–6L configuration with a total of 2880 parameters achieved an $L_{2}$ error of $1.4219\times 10^{-7}$ and $R^{2}=1.0$ after 4739 training epochs. In Case 2, the 4Q–6L configuration, consisting of 2304 parameters with the same number of layers, converged to an $L_{2}$ error of $1.4032\times 10^{-7}$ under the condition $\beta =1\times 10^{-6}$, requiring 7636 epochs for training. Under the same architecture and a penalty coefficient of $\beta =1\times 10^{-5}$, the model reached an $L_{2}$ error of $1.6932\times 10^{-7}$ in only 570 epochs. In Case 3, the 6Q–6L model with 3456 parameters converged to $L_{2}$ errors of $1.0637\times 10^{-6}$ and $1.0364\times 10^{-6}$ for penalty parameters $\beta =1\times 10^{-6}$ and $\beta =1\times 10^{-5}$, respectively, with training times of 316.643 s and 316.336 s. Lastly, in Case 4, the 5Q–6L configuration, comprising 2880 parameters, achieved an $L_{2}$ error of $6.8858\times 10^{-7}$ and $R^{2}=0.99997$ under the condition $\beta =1\times 10^{-6}$, yielding a satisfactory result.

Figure 17 and Figure 18 show the prediction results and $L_{2}$ error convergence trends for the 6-by-6 grid dome structure under the condition $\beta =1\times 10^{-6}$. In Case 1, the 5Q–6L configuration, with a total of 2880 parameters, achieved an $L_{2}$ error of $1.4219\times 10^{-7}$. Case 2, using the same architecture with 2304 parameters, reached an $L_{2}$ error of $1.4032\times 10^{-7}$, demonstrating relatively stable prediction performance. In contrast, Case 3, with a 6Q–6L configuration and 3456 parameters, converged to an $L_{2}$ error of $1.0637\times 10^{-6}$. Case 4 also yielded satisfactory results, achieving an $L_{2}$ error of $6.8858\times 10^{-7}$ with 2880 parameters.

The displacement analysis results for the 6-by-6 square grid dome structure are summarized in Table 10, while Figure 19 presents the prediction results under the condition $\beta =1\times 10^{-6}$. The $L_{2}$ error convergence trend is shown in Figure 20. The displacement control formulation described in Equation (Equation 44) of Section 4.5.3 was applied using the same parameter values as in the axial force analysis.

In Case 1, the 3Q–6L configuration with a total of 1458 parameters achieved an $L_{2}$ error of $2.5581\times 10^{-9}$ and $R^{2}=1.0$ within 762 training epochs. In Case 2, the same 3Q–6L architecture with 1458 parameters converged to an $L_{2}$ error of $1.9097\times 10^{-10}$ and $R^{2}=1.0$ in 946 epochs under the condition $\beta =1\times 10^{-6}$. In contrast, under the condition $\beta =1\times 10^{-5}$, the model with 2430 parameters reached an $L_{2}$ error of $4.6265\times 10^{-9}$ after 9674 epochs. In Case 3, the 6Q–6L configuration with 2916 parameters achieved an $L_{2}$ error of $2.5805\times 10^{-9}$ in 662 epochs under $\beta =1\times 10^{-6}$. Conversely, under $\beta =1\times 10^{-5}$, the 4Q–6L model with 1944 parameters converged to an $L_{2}$ error of $4.3679\times 10^{-8}$ in 5633 epochs. In Case 4, the 6Q–6L configuration with 2916 parameters achieved an $L_{2}$ error of $3.4688\times 10^{-9}$ and $R^{2}=1.0$ in 659 epochs under $\beta =1\times 10^{-6}$. Under the same condition, the 3Q–6L configuration with 1458 parameters reached an $L_{2}$ error of $1.0672\times 10^{-7}$ at 1103 epochs, yielding reasonably satisfactory results.

### 5.4. Discussion

This study performed structural analyses on various truss systems by integrating an index-based quantum neural network (QNN) architecture with a mechanics-informed loss function. The predictive performance was compared with that of a conventional neural network (NN), and convergence characteristics were quantitatively analyzed depending on structural types and hyperparameter settings.

In the case of the 10-bar planar truss (Case 1), the QNN produced lower $L_{2}$ errors with fewer parameters compared to the NN. For example, the 4Q–6L QNN configuration achieved an $L_{2}$ error of $1.9314\times 10^{-10}$ while using approximately 64% of the parameters required by the NN. In some configurations, the QNN yielded acceptable results with less than half the number of parameters.

To examine the impact of the proposed indexed QNN architecture settings, a grid search was conducted over 2–11 qubits and 2–11 layers, and the distribution of the coefficient of determination ($R^{2}$) was evaluated. The results indicated that training performance remained stable around 6 layers, while configurations with more than 7 qubits showed decreasing predictive accuracy as the number of layers increased. Accordingly, the training for Cases 2 through 4 was limited to 2–6 qubits and 2–6 layers. The domain decomposition method also showed a reduction in parameter count.

Similar patterns were observed in the analysis of the 25-bar truss. In Case 2, the single-domain 2Q–16L model required 96 parameters and 822 epochs to reach an $L_{2}$ error of $2.4883\times 10^{-12}$, whereas the domain-split 2Q–3L model reached an $L_{2}$ error of $3.4959\times 10^{-14}$ in 533 epochs with 90 parameters. In Case 3, the single-domain 3Q–18L model reached an $L_{2}$ error of $3.6791\times 10^{-8}$ using 162 parameters and 6027 epochs, while the domain-split 2Q–4L model achieved $2.6853\times 10^{-9}$ in 2615 epochs with 120 parameters. In Case 4, the single-domain 3Q–19L model required 171 parameters and 2421 epochs to achieve $2.9086\times 10^{-8}$, whereas the domain-split 2Q–5L model reached $1.8233\times 10^{-9}$ in 183 epochs with 150 parameters.

The domain decomposition method was also applied to the 25-bar space truss. In both Case 1 and Case 2, the same 2Q–3L configuration was used, and the models converged to $L_{2}$ errors of $3.4760\times 10^{-13}$ and $2.9885\times 10^{-13}$ at 1208 and 1236 epochs, respectively, each with 90 parameters. In Case 3, the 2Q–6L model reached $7.4887\times 10^{-10}$ in 1630 epochs using 180 parameters. In Case 4, the 3Q–5L model achieved $3.9648\times 10^{-10}$ in 9478 epochs with 225 parameters, indicating acceptable prediction performance.

Axial force and displacement analysis was also conducted for the 6×6 grid dome structure. In Case 1, the 5Q–6L model converged to an $L_{2}$ error of $1.4219\times 10^{-7}$. In Case 2, the 4Q–6L model under the condition $\beta =1\times 10^{-6}$ achieved an error of $1.4032\times 10^{-7}$, and, under $\beta =1\times 10^{-5}$, the same model reached $1.6932\times 10^{-7}$ in 570 epochs. In Case 3 and Case 4, the 6Q–6L and 5Q–6L models, respectively, converged to errors on the order of $10^{-6}$, indicating that the QNN framework was applicable, even to complex structural configurations.

These results show that the domain decomposition method reduced the total number of parameters by approximately 6.25% to 25% while also yielding acceptable results in terms of prediction accuracy and convergence speed compared to single-domain models. The experiments demonstrated that the indexed QNN based on domain decomposition produced valid results in the structural analysis.

## 6. Conclusions

In this study, an index-based quantum neural network (QNN) based on a variational quantum circuit (VQC) is proposed as a surrogate model for the analysis of axial force and displacement in truss structures. The proposed QNN adopts a discrete input format based on member and node indices instead of coordinate-based representations, and it also introduces a domain-separate strategy. The loss function includes physics-based constraints derived from the force method and displacement method, and it is formulated using an augmented Lagrangian approach to naturally incorporate structural mechanics information into the training process.

Numerical experiments were conducted on various structural configurations, including the 10-bar and 25-bar planar and spatial trusses and the 6 × 6 grid dome. For relatively simple structures, stable learning was achieved through shallow circuit configurations, and, as the structural complexity increased, the domain-separate strategy showed advantageous performance in terms of parameter efficiency. In particular, the domain-separate strategy was able to achieve satisfactory results, even with the same or fewer number of parameters compared to the single-domain configuration.

According to the performance analysis based on the number of qubits and circuit depth, the most stable convergence tendency appeared around six layers, and increasing the number of qubits did not always lead to improved performance. This suggests that, when applying QNNs to structural analysis problems, circuit expansion alone is not sufficient, and modeling strategies, such as domain separation, must also be applied together.

In conclusion, this study numerically validated the practical applicability of QNN-based structural analysis even when under limited quantum resources. Future research directions include the design of quantum circuits robust to noise, the optimization of adaptive ansatz structures, and the application of quantum natural gradient-based learning methods. These approaches are expected to contribute to expanding QNN-based numerical analysis to advanced structural applications, such as real-time monitoring, damage detection, and optimal design.

Furthermore, in order to enhance the performance of QNNs and perform more precise structural analysis, an in-depth investigation into the role of quantum entanglement is necessary. Although this study used quantum circuits with strong entanglement, future studies should systematically analyze the effects of entanglement strength, depth, and configuration on prediction accuracy and convergence characteristics. In particular, theoretical and numerical validation of the synergy between domain separation and entanglement structure could provide insights into the efficient learning of complex structural interactions and enhance the practicality and scalability of QNN-based structural analysis.

## Figures and Tables

**Figure 1 biomimetics-10-00407-f001:**
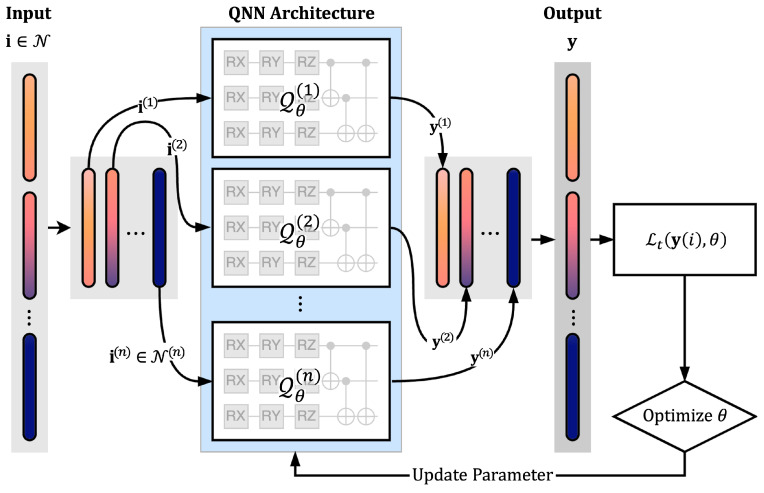
Flowchart of the proposed structural analysis process using QNN.

**Figure 2 biomimetics-10-00407-f002:**
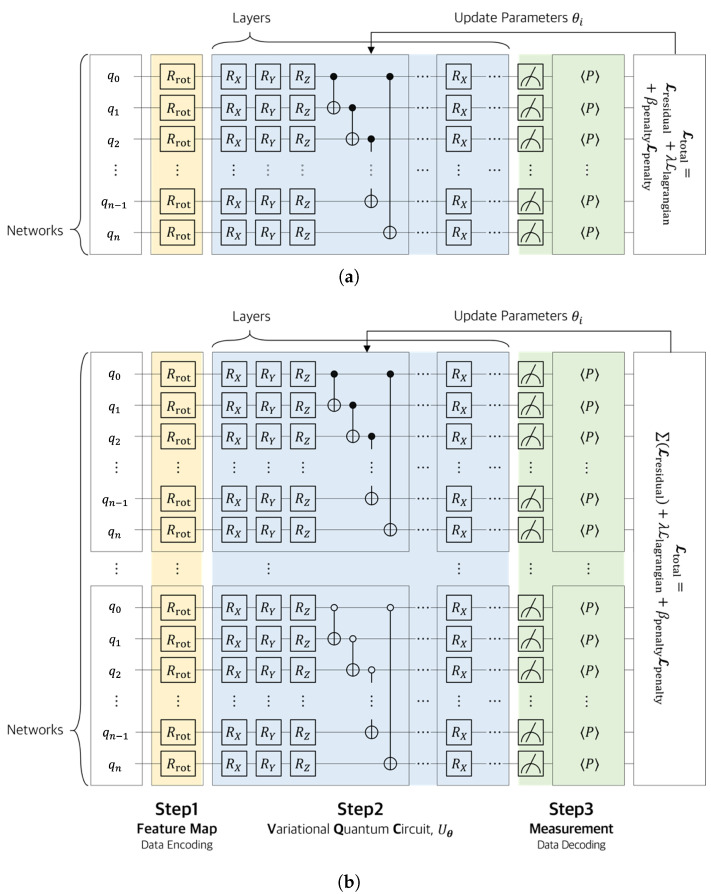
Two types of quantum neural network models: (**a**) single-domain model using a single quantum circuit (VQC); and (**b**) multi-domain model that separates the input into multiple domains, each processed in parallel by an independent quantum circuit.

**Figure 3 biomimetics-10-00407-f003:**
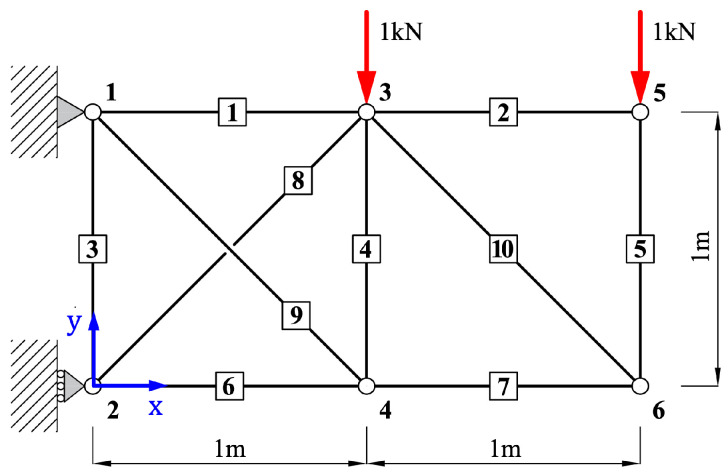
Configurations of a 10-bar plane truss.

**Figure 4 biomimetics-10-00407-f004:**
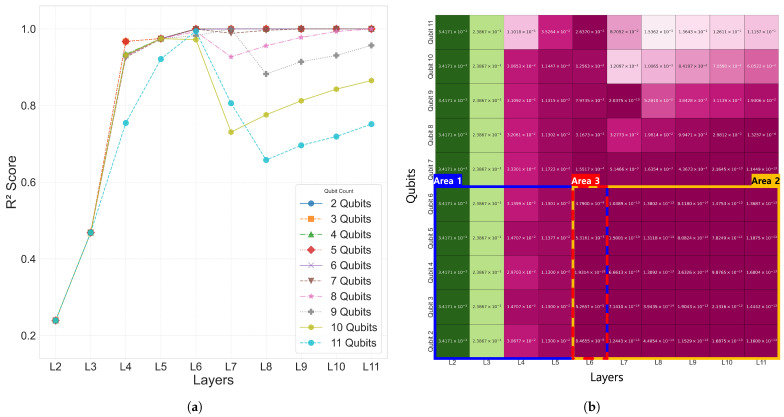
Performance evaluation of the QNN models with 2 to 11 qubits and layers. (**a**) $R^{2}$ score distribution for Case 1. (**b**) $L_{2}$ error for varying qubit and layer configurations.

**Figure 5 biomimetics-10-00407-f005:**
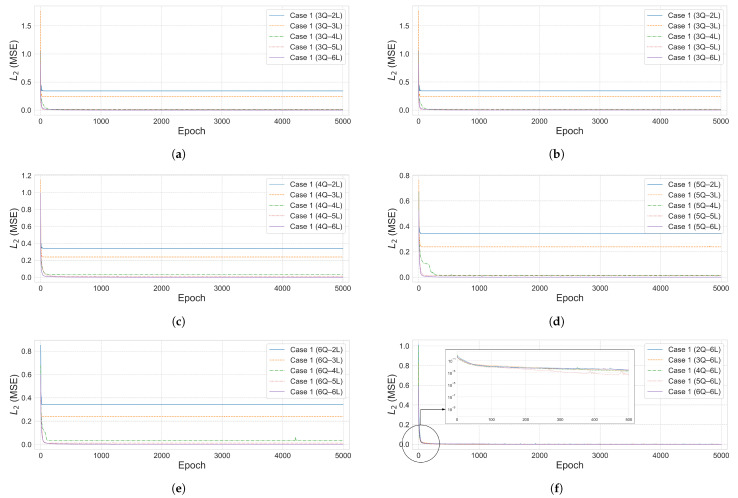
$L_{2}$ error curves for the 10-bar plane truss at $\beta =1\times \;10^{-2}$. (**a**–**e**) Qubit–layer combinations within Area 1 (2–6 qubits, 2–6 layers). (**f**) Fixed 6-layer configurations across 2–6 qubits at the minimum $L_{2}$ epoch.

**Figure 6 biomimetics-10-00407-f006:**
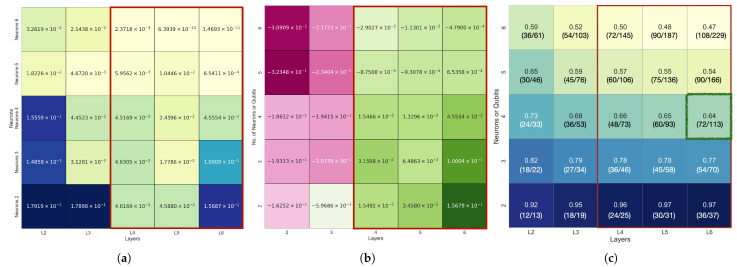
Comparison with previous studies: (**a**) $L_{2}$ error of the neural network model ($L_{2}^{\left(D\right)}$), (**b**) error difference between classical and quantum models ($L_{2}^{\left(D\right)}-L_{2}^{\left(Q\right)}$), and (**c**) the parameter ratio between quantum and neural network models ($n^{\left(Q\right)}/n^{\left(N\right)}$).

**Figure 7 biomimetics-10-00407-f007:**
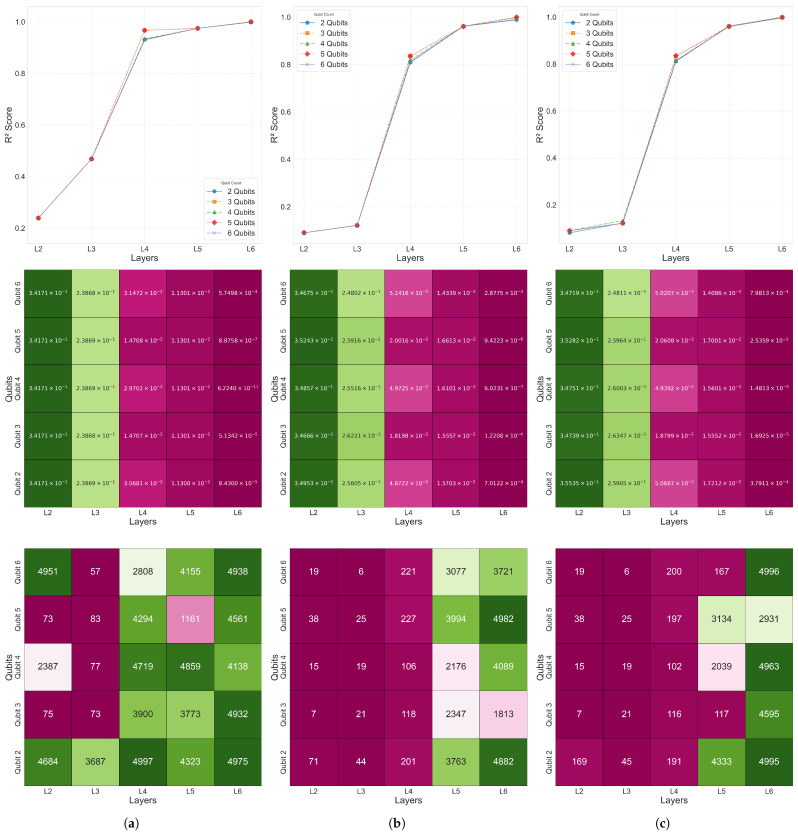
The 10-bar plane truss $R^{2}$, MSE, and minimum $L_{2}$ error epoch heatmaps for $\beta =1\times \;10^{-2}$: (**a**) Case 2, (**b**) Case 3, (**c**) and Case 4.

**Figure 8 biomimetics-10-00407-f008:**
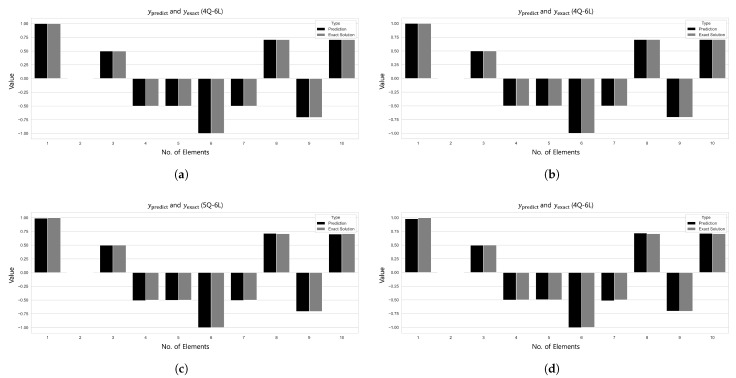
Comparison of $y_{\mathrm{pred}}$ and $y_{\mathrm{exact}}$ for $\beta =1\times \;10^{-2}$: (**a**) Case 1: 4Q–6L; (**b**) Case 2: 4Q–6L; (**c**) Case 3: 5Q–6L; and (**d**) Case 4: 4Q–6L.

**Figure 9 biomimetics-10-00407-f009:**
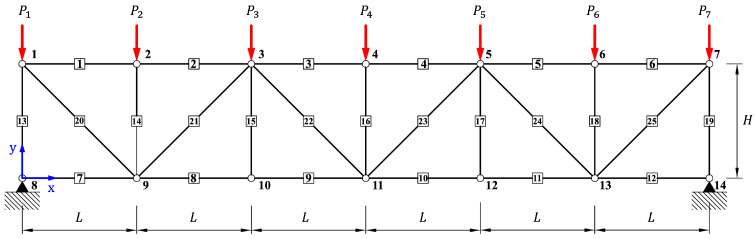
Configurations of a 25-bar plane truss.

**Figure 10 biomimetics-10-00407-f010:**
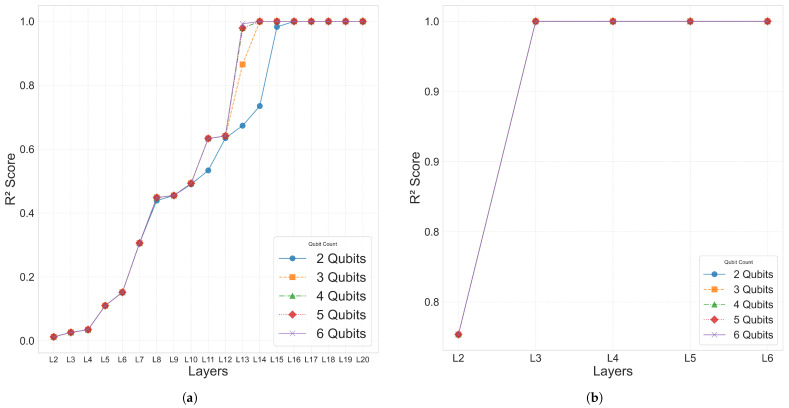
$R^{2}$ distributions for a QNN-based analysis of the 25-bar truss (Case 1): (**a**) Single-domain model with 2–11 qubits and 2–11 layers. (**b**) Five-domain model with 2–6 qubits and 2–6 layers.

**Figure 11 biomimetics-10-00407-f011:**
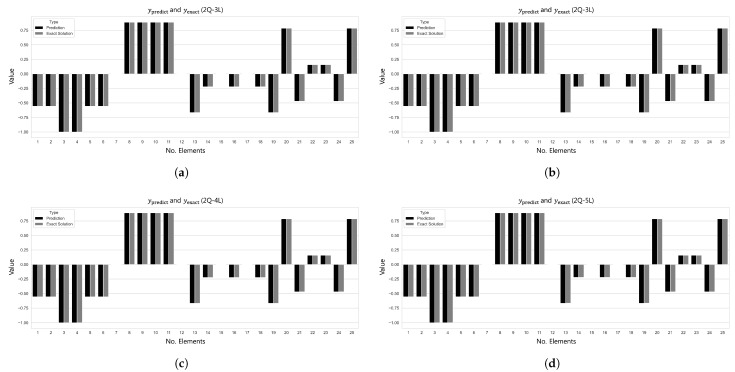
Comparison of $y_{\mathrm{pred}}$ and $y_{\mathrm{exact}}$ for $\beta =1\times 10^{6}$: (**a**) Case 1: 5 Domain-2Q-3L, (**b**) Case 2: 5 Domain-2Q-3L, (**c**) Case 3: 5 Domain-2Q-4L, and (**d**) Case 4: 5 Domain-2Q-5L.

**Figure 12 biomimetics-10-00407-f012:**
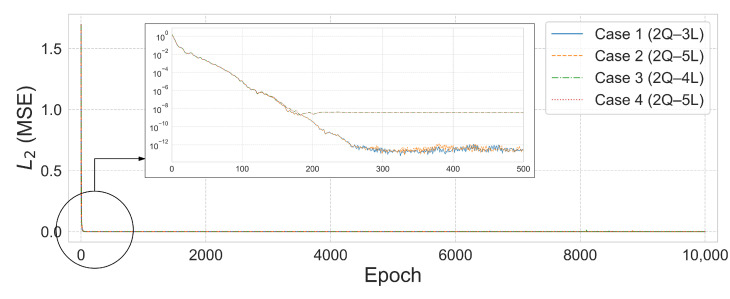
$L_{2}$ error comparison graph for the 25-bar truss with 5 domains, $\beta =1\;\times \;10^{-6}$.

**Figure 13 biomimetics-10-00407-f013:**
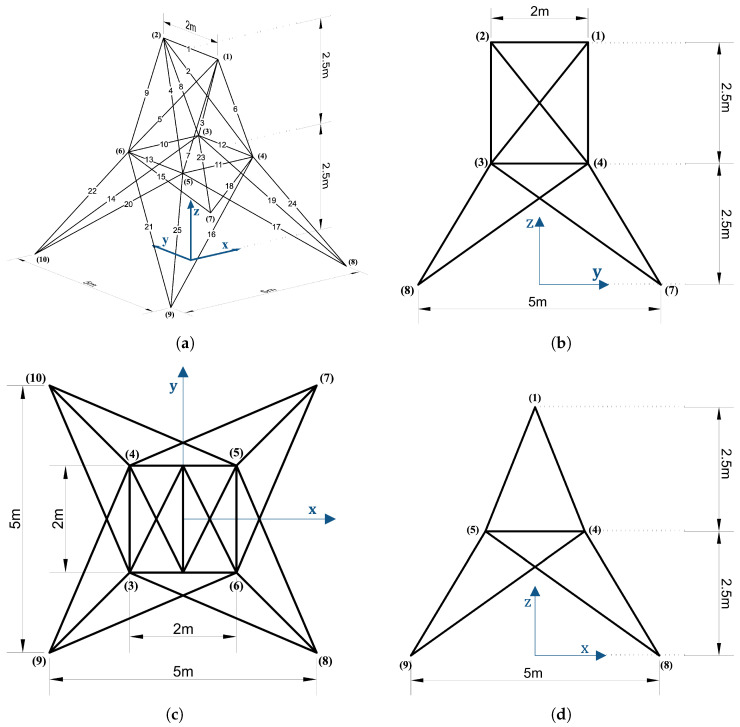
Configurations of the 25-bar space truss: (**a**) 3D view; (**b**) side view in the *y*–*z* plane; (**c**) side view in the *x*–*z* plane; and (**d**) top view.

**Figure 14 biomimetics-10-00407-f014:**
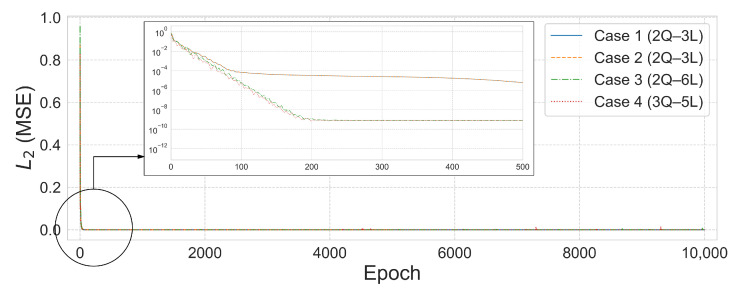
$L_{2}$ error comparison graph for the 25-bar space truss with 5 domains, $\beta =1\;\times \;10^{-6}$.

**Figure 15 biomimetics-10-00407-f015:**
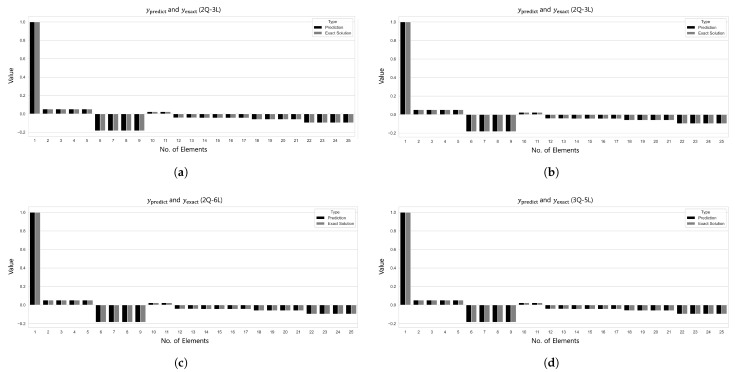
Comparison of $y_{\mathrm{pred}}$ and $y_{\mathrm{exact}}$ for $\beta =1\times 10^{-6}$: (**a**) Case 1: 5 Domain-2Q-3L, (**b**) Case 2: 5 Domain-2Q-3L, (**c**) Case 3: 5 Domain-2Q-6L, and (**d**) Case 4: 5 Domain-3Q-5L.

**Figure 16 biomimetics-10-00407-f016:**
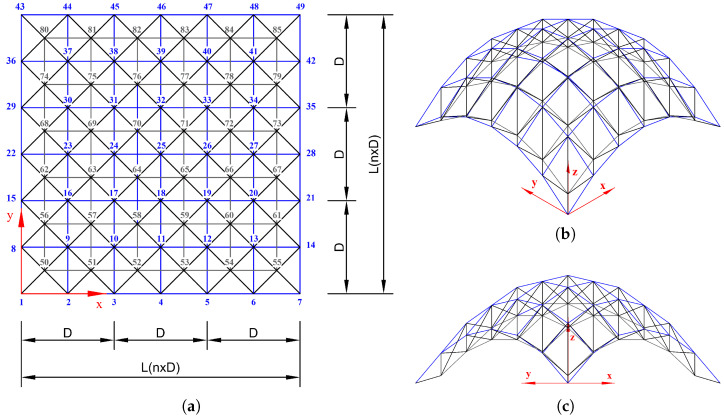
Geometry of the 6-by-6 square grid dome. (**a**) Plan view of the *x*–*y* plane, and (**b**,**c**) perspective views.

**Figure 17 biomimetics-10-00407-f017:**
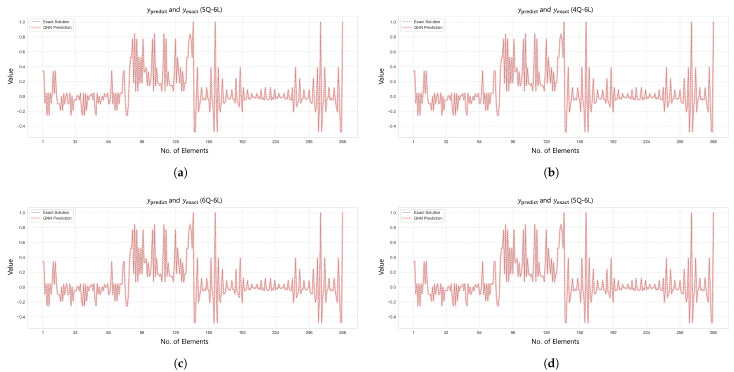
Comparison of $y_{\mathrm{pred}}$ and $y_{\mathrm{exact}}$ for $\beta =1\;\times \;10^{6}$: (**a**) Case 1: 32 Domain-5Q-6L, (**b**) Case 2: 32 Domain-4Q-6L, (**c**) Case 3: 32 Domain-6Q-6L, and (**d**) Case 4: 32 Domain-5Q-6L.

**Figure 18 biomimetics-10-00407-f018:**
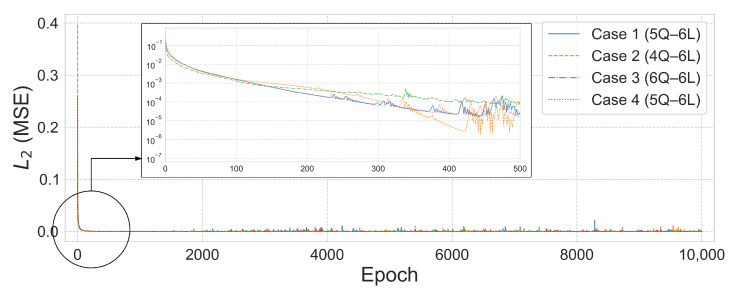
$L_{2}$ error comparison graph for the 6-by-6 square grid dome with 32 domains, $\beta =1\;\times \;10^{-6}$.

**Figure 19 biomimetics-10-00407-f019:**
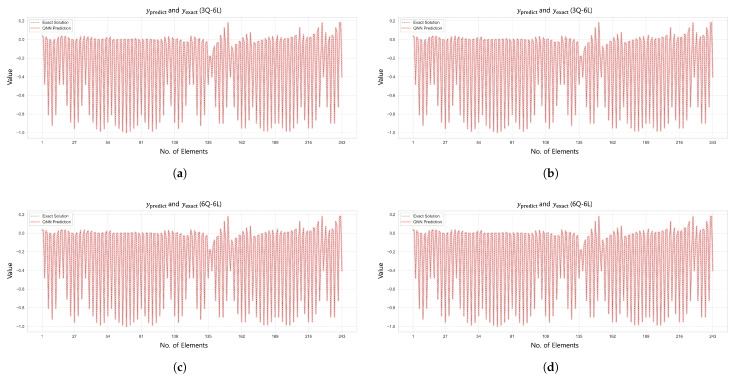
Comparison of QNN performance for $\beta =1\;\times \;10^{6}$: (**a**) Case 1: 27 Domain-3Q-6L, (**b**) Case 2: 27 Domain-3Q-6L, (**c**) Case 3: 27 Domain-6Q-6L, and (**d**) Case 4: 27 Domain-6Q-6L.

**Figure 20 biomimetics-10-00407-f020:**
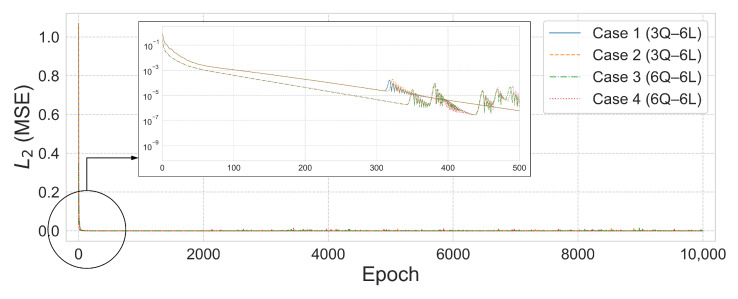
$L_{2}$ error comparison graph for the 6-by-6 square grid dome with 27 domains, $\beta =1\;\times \;10^{-6}$.

**Table 1 biomimetics-10-00407-t001:** Loss function configurations for QNN training.

Case	Loss Function Expression
Case 1 (Residual Only)	$L_{r}$
Case 2 (Quadratic Penalty)	$L_{r}+\beta L_{p}$
Case 3 (Lagrangian Only) *	$L_{r}+\lambda^{T}L_{p}$
Case 4 (Augmented Lagrangian) *	$L_{r}+\lambda^{T}L_{p}+\beta L_{p}$

* In Case 3, the Lagrangian multiplier λ was updated based on a learning rate η, while, in Case 4, λ was updated based on the penalty parameter β (see Equations (Equation 41) and (Equation 43)).

**Table 2 biomimetics-10-00407-t002:** Summary of the truss and dome example models used for structural analysis.

Structure Type	Nodes	Elements	Problem
10-bar Plane Truss	6	10	Force
25-bar Plane Truss	14	25	Force
25-bar Space Truss	10	25	Force
6-by-6 Square Grid Dome	85	288 *	Force and Displacement

* For the displacement prediction model, the number of degrees of freedom (DOF) was 243.

**Table 3 biomimetics-10-00407-t003:** Case 1—10-bar plane truss: detailed comparative results.

Qubits	Layers	Total Params	Epoch	$L_{2}$ Error	$|L_{t}|$	$R^{2}$ Score	Time (s)
2	2	12	301	3.4171 $\times \;10^{-1}$	2.3920 $\times \;10^{-1}$	4.6522 $\times \;10^{-1}$	1.28913
2	3	18	635	2.3867 $\times \;10^{-1}$	4.6861 $\times \;10^{-1}$	4.9101 $\times \;10^{-1}$	1.16465
2	4	24	4990	3.0677 $\times \;10^{-2}$	9.3170 $\times \;10^{-1}$	6.0451 $\times \;10^{-2}$	1.32921
2	5	30	4975	1.1300 $\times \;10^{-2}$	9.7484 $\times \;10^{-1}$	2.0962 $\times \;10^{-2}$	1.44017
2	6	36	5000	8.4655 $\times \;10^{-5}$	9.9981 $\times \;10^{-1}$	2.4227 $\times \;10^{-4}$	1.63554
3	2	18	269	3.4171 $\times \;10^{-1}$	1.0136 $\times \;10^{0}$	2.3920 $\times \;10^{-1}$	1.58692
3	3	27	2678	2.3867 $\times \;10^{-1}$	6.8796 $\times \;10^{-1}$	4.6861 $\times \;10^{-1}$	1.41797
3	4	36	515	1.4707 $\times \;10^{-2}$	2.2248 $\times \;10^{-1}$	9.6726 $\times \;10^{-1}$	1.62781
3	5	45	1549	1.1300 $\times \;10^{-2}$	1.6334 $\times \;10^{-1}$	9.7484 $\times \;10^{-1}$	2.07655
3	6	54	3902	5.2657 $\times \;10^{-5}$	1.1168 $\times \;10^{-2}$	9.9988 $\times \;10^{-1}$	2.31483
4	2	24	655	3.4171 $\times \;10^{-1}$	1.0136 $\times \;10^{0}$	2.3920 $\times \;10^{-1}$	1.54701
4	3	36	1274	2.3867 $\times \;10^{-1}$	6.8796 $\times \;10^{-1}$	4.6861 $\times \;10^{-1}$	2.14466
4	4	48	4871	2.9703 $\times \;10^{-2}$	2.7311 $\times \;10^{-1}$	9.3387 $\times \;10^{-1}$	1.90128
4	5	60	4243	1.1300 $\times \;10^{-2}$	1.6331 $\times \;10^{-1}$	9.7484 $\times \;10^{-1}$	2.26965
4	6	72	4532	1.9314 $\times \;10^{-10}$	2.7657 $\times \;10^{-5}$	1.0000 $\times \;10^{0}$	3.07850
5	2	30	1971	3.4171 $\times \;10^{-1}$	1.0136 $\times \;10^{0}$	2.3920 $\times \;10^{-1}$	1.78833
5	3	45	394	2.3867 $\times \;10^{-1}$	6.8796 $\times \;10^{-1}$	4.6861 $\times \;10^{-1}$	1.98362
5	4	60	4352	1.4707 $\times \;10^{-2}$	2.2243 $\times \;10^{-1}$	9.6726 $\times \;10^{-1}$	2.41754
5	5	75	4592	1.1377 $\times \;10^{-2}$	1.7084 $\times \;10^{-1}$	9.7467 $\times \;10^{-1}$	2.73436
5	6	90	4444	5.3161 $\times \;10^{-7}$	1.1944 $\times \;10^{-3}$	1.0000 $\times \;10^{0}$	4.30919
6	2	36	2591	3.4171 $\times \;10^{-1}$	1.0137 $\times \;10^{0}$	2.3920 $\times \;10^{-1}$	2.13546
6	3	54	2527	2.3867 $\times \;10^{-1}$	6.8802 $\times \;10^{-1}$	4.6861 $\times \;10^{-1}$	2.34046
6	4	72	4976	3.1399 $\times \;10^{-2}$	2.8509 $\times \;10^{-1}$	9.3009 $\times \;10^{-1}$	2.87620
6	5	90	4288	1.1301 $\times \;10^{-2}$	1.6325 $\times \;10^{-1}$	9.7484 $\times \;10^{-1}$	5.02538
6	6	108	4975	4.7900 $\times \;10^{-4}$	3.4637 $\times \;10^{-2}$	9.9893 $\times \;10^{-1}$	4.10551

**Table 4 biomimetics-10-00407-t004:** Case 1—25-bar plane truss: detailed comparative results.

Domain = 1 (Single Domain)							
Qubits	Layers	Total Params	Epoch	$L_{2}$ Error	$|L_{t}|$	$R^{2}$ Score	Time (s)
2	16	96	1154	2.5613 $\times \;10^{-12}$	2.5613 $\times \;10^{-12}$	1.0000 $\times \;10^{0}$	5.40458
3	14	126	6193	4.2138 $\times \;10^{-11}$	4.2138 $\times \;10^{-11}$	1.0000 $\times \;10^{0}$	6.94795
4	14	168	716	2.8528 $\times \;10^{-13}$	2.8528 $\times \;10^{-13}$	1.0000 $\times \;10^{0}$	8.36999
5	14	210	6454	3.2832 $\times \;10^{0}$	3.2832 $\times \;10^{-13}$	1.0000 $\times \;10^{0}$	15.98381
6	14	252	5977	4.0991 $\times \;10^{-13}$	4.0991 $\times \;10^{-13}$	1.0000 $\times \;10^{0}$	19.66124
**Domain = 5 (Separated Domain)**							
**Qubits**	**Layers**	**Total Params**	**Epoch**	**$L_{2}$ Error**	** $|L_{t}|$ **	**$R^{2}$ Score**	**Time (s)**
2	3	90	526	2.8102 $\times \;10^{-14}$	2.9802 $\times \;10^{-07}$	1.0000 $\times \;10^{0}$	5.08589
3	3	135	520	2.8386 $\times \;10^{-14}$	3.5763 $\times \;10^{-7}$	1.0000 $\times \;10^{0}$	6.77821
4	3	180	350	1.8270 $\times \;10^{-14}$	2.9802 $\times \;10^{-7}$	1.0000 $\times \;10^{0}$	7.56486
5	3	225	310	3.4150 $\times \;10^{-14}$	4.4703 $\times \;10^{-7}$	1.0000 $\times \;10^{0}$	10.51405
6	3	270	319	4.0323 $\times \;10^{-14}$	4.7684 $\times \;10^{-7}$	1.0000 $\times \;10^{0}$	15.34173

**Table 5 biomimetics-10-00407-t005:** Case 2—25-bar plane truss: detailed comparative results.

Domain = 1 (Single Domain)								
** β **	Qubits	Layers	Total Params	Epoch	$L_{2}$ Error	$|L_{t}|$	$R^{2}$ Score	Time (s)
1 $\times \;10^{-6}$	2	16	96	822	2.4883 $\times \;10^{-12}$	3.4959 $\times \;10^{-14}$	1.0000 $\times \;10^{0}$	5.02629
1 $\times \;10^{-5}$	2	16	96	788	4.3938 $\times \;10^{-12}$	4.3939 $\times \;10^{-12}$	1.0000 $\times \;10^{0}$	5.89781
1 $\times \;10^{-2}$	2	16	96	822	2.4883 $\times \;10^{-12}$	2.5561 $\times \;10^{-12}$	1.0000 $\times \;10^{0}$	5.83060
**Domain = 5 (Separated Domain)**								
** β **	**Qubits**	**Layers**	**Total Params**	**Epoch**	**$L_{2}$ Error**	** $|L_{t}|$ **	**$R^{2}$ Score**	**Time (s)**
1 $\times \;10^{-6}$	2	3	90	533	3.4959 $\times \;10^{-14}$	3.4959 $\times \;10^{-14}$	1.0000 $\times \;10^{0}$	5.02629
1 $\times \;10^{-5}$	2	3	90	359	3.9277 $\times \;10^{-14}$	3.9278 $\times \;10^{-14}$	1.0000 $\times \;10^{0}$	5.20799
1 $\times \;10^{-2}$	2	3	90	386	2.9523 $\times \;10^{-14}$	3.0109 $\times \;10^{-14}$	1.0000 $\times \;10^{0}$	5.11479

**Table 6 biomimetics-10-00407-t006:** Case 3—25-bar plane truss: detailed comparative results.

Domain = 1 (Single Domain)								
** β **	Qubits	Layers	Total Params	Epoch	$L_{2}$ Error	$|L_{t}|$	$R^{2}$ Score	Time (s)
1 $\times \;10^{-6}$	3	18	162	6027	3.6791 $\times \;10^{-8}$	4.5719 $\times \;10^{-8}$	1.0000 $\times \;10^{0}$	8.961
1 $\times \;10^{-5}$	3	19	171	9950	4.4570 $\times \;10^{-7}$	4.5053 $\times \;10^{-7}$	1.0000 $\times \;10^{0}$	8.533
1 $\times \;10^{-2}$	6	20	360	4937	2.1295 $\times \;10^{-2}$	8.3458 $\times \;10^{-3}$	8.0375 $\times \;10^{-1}$	30.319
**Domain = 5 (Separated Domain)**								
** β **	**Qubits**	**Layers**	**Total Params**	**Epoch**	**$L_{2}$ Error**	** $|L_{t}|$ **	**$R^{2}$ Score**	**Time (s)**
1 $\times \;10^{-6}$	2	4	120	2615	2.6853 $\times \;10^{-9}$	3.0108 $\times \;10^{-9}$	1.0000 $\times \;10^{0}$	6.66417
1 $\times \;10^{-5}$	3	4	180	8370	5.1807 $\times \;10^{-9}$	2.9732 $\times \;10^{-9}$	1.0000 $\times \;10^{0}$	9.19804
1 $\times \;10^{-2}$	6	5	450	3841	1.6800 $\times \;10^{-2}$	3.4354 $\times \;10^{-2}$	7.2672 $\times \;10^{-1}$	24.66548

**Table 7 biomimetics-10-00407-t007:** Case 4—25-bar plane truss: detailed comparative results.

Domain = 1 (Single Domain)								
** β **	Qubits	Layers	Total Params	Epoch	$L_{2}$ Error	$|L_{t}|$	$R^{2}$ Score	Time (s)
1 $\times \;10^{-6}$	3	19	171	2421	2.9086 $\times \;10^{-8}$	2.2918 $\times \;10^{-8}$	1.0000 $\times \;10^{0}$	7.838
1 $\times \;10^{-5}$	4	19	228	7435	5.7997 $\times \;10^{-7}$	5.1730 $\times \;10^{-7}$	1.0000 $\times \;10^{0}$	11.927
1 $\times \;10^{-2}$	6	19	342	9803	2.1660 $\times \;10^{-2}$	3.2157 $\times \;10^{-2}$	8.3604 $\times \;10^{-1}$	29.565
**Domain = 5 (Separated Domain)**								
** β **	**Qubits**	**Layers**	**Total Params**	**Epoch**	**$L_{2}$ Error**	** $|L_{t}|$ **	**$R^{2}$ Score**	**Time (s)**
1 $\times \;10^{-6}$	2	5	150	183	1.8233 $\times \;10^{-9}$	1.7689 $\times \;10^{-9}$	1.0000 $\times \;10^{0}$	7.11437
1 $\times \;10^{-5}$	5	5	375	9803	1.4527 $\times \;10^{-9}$	1.5128 $\times \;10^{-9}$	9.9998 $\times \;10^{-1}$	18.35036
1 $\times \;10^{-2}$	6	4	360	6368	1.4292 $\times \;10^{-2}$	3.3215 $\times \;10^{-2}$	6.8316 $\times \;10^{-1}$	18.32772

**Table 8 biomimetics-10-00407-t008:** Case 1–4—25-bar space truss: detailed comparative results.

Case 1; Domain = 5 (Separated Domain)								
** β **	Qubits	Layers	Total Params	Epoch	$L_{2}$ Error	$|L_{t}|$	$R^{2}$ Score	Time (s)
–	2	3	90	1208	3.4760 $\times \;10^{-13}$	3.4760 $\times \;10^{-13}$	1.0000 $\times \;10^{0}$	4.946
**Case 2; Domain = 5 (Separated Domain)**								
** β **	**Qubits**	**Layers**	**Total Params**	**Epoch**	**$L_{2}$ Error**	** $|L_{t}|$ **	**$R^{2}$ Score**	**Time (s)**
1 $\times \;10^{-6}$	2	3	90	1236	2.9885 $\times \;10^{-13}$	2.9885 $\times \;10^{-13}$	1.0000 $\times \;10^{0}$	3.650
1 $\times \;10^{-5}$	2	3	90	1975	2.7029 $\times \;10^{-13}$	2.7029 $\times \;10^{-13}$	1.0000 $\times \;10^{0}$	3.553
1 $\times \;10^{-2}$	2	3	90	1369	8.5238 $\times \;10^{-12}$	8.7200 $\times \;10^{-12}$	1.0000 $\times \;10^{0}$	3.979
**Case 3; Domain = 5 (Separated Domain)**								
** β **	**Qubits**	**Layers**	**Total Params**	**Epoch**	**$L_{2}$ Error**	** $|L_{t}|$ **	**$R^{2}$ Score**	**Time (s)**
1 $\times \;10^{-6}$	2	6	180	1630	7.4887 $\times \;10^{-10}$	7.3686 $\times \;10^{-10}$	1.0000 $\times \;10^{0}$	9.586
1 $\times \;10^{-5}$	3	6	270	9753	1.3174 $\times \;10^{-9}$	1.3358 $\times \;10^{-9}$	1.0000 $\times \;10^{0}$	13.595
1 $\times \;10^{-2}$	6	4	360	11	7.4059 $\times \;10^{-3}$	1.1727 $\times \;10^{-2}$	7.4215 $\times \;10^{-1}$	19.265
**Case 4; Domain = 5 (Separated Domain)**								
** β **	**Qubits**	**Layers**	**Total Params**	**Epoch**	**$L_{2}$ Error**	** $|L_{t}|$ **	**$R^{2}$ Score**	**Time (s)**
1 $\times \;10^{-6}$	3	5	225	9478	3.9648 $\times \;10^{-10}$	4.1852 $\times \;10^{-10}$	1.0000 $\times \;10^{0}$	9.269
1 $\times \;10^{-5}$	5	5	375	5802	3.0698 $\times \;10^{-9}$	3.1014 $\times \;10^{-9}$	1.0000 $\times \;10^{0}$	14.580
1 $\times \;10^{-2}$	6	4	360	10	9.9744 $\times \;10^{-3}$	2.9379 $\times \;10^{-2}$	7.2720 $\times \;10^{-1}$	16.755

**Table 9 biomimetics-10-00407-t009:** Case 1–4—6-by-6 grid dome: axial force prediction results.

Case 1; Domain = 32 (Separated Domain)								
** β **	Qubits	Layers	Total Params	Epoch	$L_{2}$ Error	$|L_{t}|$	$R^{2}$ Score	Time (s)
–	5	6	2880	4739	1.4219 $\times \;10^{-7}$	1.4219 $\times \;10^{-7}$	1.0000 $\times \;10^{0}$	196.283
**Case 2; Domain = 32 (Separated Domain)**								
** β **	**Qubits**	**Layers**	**Total Params**	**Epoch**	**$L_{2}$ Error**	** $|L_{t}|$ **	**$R^{2}$ Score**	**Time (s)**
1 $\times \;10^{-6}$	4	6	2304	7636	1.4032 $\times \;10^{-7}$	1.4032 $\times \;10^{-7}$	1.0000 $\times \;10^{0}$	118.481
1 $\times \;10^{-5}$	4	6	2304	570	1.6932 $\times \;10^{-7}$	1.6932 $\times \;10^{-7}$	1.0000 $\times \;10^{0}$	115.218
**Case 3; Domain = 32 (Separated Domain)**								
** β←η **	**Qubits**	**Layers**	**Total Params**	**Epoch**	**$L_{2}$ Error**	** $|L_{t}|$ **	**$R^{2}$ Score**	**Time (s)**
1 $\times \;10^{-6}$	6	6	3456	4599	1.0637 $\times \;10^{-6}$	6.3044 $\times \;10^{-7}$	9.9996 $\times \;10^{-1}$	316.643
1 $\times \;10^{-5}$	6	6	3456	8198	1.0364 $\times \;10^{-6}$	1.3578 $\times \;10^{-7}$	9.9965 $\times \;10^{-1}$	316.336
**Case 4; Domain = 32 (Separated Domain)**								
** β **	**Qubits**	**Layers**	**Total Params**	**Epoch**	**$L_{2}$ Error**	** $|L_{t}|$ **	**$R^{2}$ Score**	**Time (s)**
1 $\times \;10^{-6}$	5	6	2880	6809	6.8858 $\times \;10^{-7}$	2.2918 $\times \;10^{-7}$	9.9997 $\times \;10^{-1}$	164.747
1 $\times \;10^{-5}$	5	6	2880	4818	1.6210 $\times \;10^{-6}$	3.7537 $\times \;10^{-7}$	9.9962 $\times \;10^{-1}$	179.735

**Table 10 biomimetics-10-00407-t010:** Case 1–4—6-by-6 grid dome: displacement prediction results.

Case 1; Domain = 27 (Separated Domain)								
** β **	Qubits	Layers	Total Params	Epoch	$L_{2}$ Error	$|L_{t}|$	$R^{2}$ Score	Time (s)
–	3	6	1458	762	2.5581 $\times \;10^{-9}$	6.9068 $\times \;10^{-8}$	1.0000 $\times \;10^{0}$	86.564
**Case 2; Domain = 27 (Separated Domain)**								
** β **	**Qubits**	**Layers**	**Total Params**	**Epoch**	**$L_{2}$ Error**	** $|L_{t}|$ **	**$R^{2}$ Score**	**Time (s)**
1 $\times \;10^{-6}$	3	6	1458	946	1.9097 $\times \;10^{-10}$	5.1562 $\times \;10^{-9}$	1.0000 $\times \;10^{0}$	68.784
1 $\times \;10^{-5}$	5	6	2430	9674	4.6265 $\times \;10^{-9}$	1.2491 $\times \;10^{-7}$	1.0000 $\times \;10^{0}$	198.522
**Case 3; Domain = 27 (Separated Domain)**								
** β←η **	**Qubits**	**Layers**	**Total Params**	**Epoch**	**$L_{2}$ Error**	** $|L_{t}|$ **	**$R^{2}$ Score**	**Time (s)**
1 $\times \;10^{-6}$	6	6	2916	662	2.5805 $\times \;10^{-9}$	5.2583 $\times \;10^{-8}$	1.0000 $\times \;10^{0}$	254.766
1 $\times \;10^{-5}$	4	6	1944	5633	4.3679 $\times \;10^{-8}$	7.2066 $\times \;10^{-7}$	1.0000 $\times \;10^{0}$	127.549
**Case 4; Domain = 27 (Separated Domain)**								
** β **	**Qubits**	**Layers**	**Total Params**	**Epoch**	**$L_{2}$ Error**	** $|L_{t}|$ **	**$R^{2}$ Score**	**Time (s)**
1 $\times \;10^{-6}$	6	6	2916	659	3.4688 $\times \;10^{-9}$	8.0135 $\times \;10^{-8}$	1.0000 $\times \;10^{0}$	197.081
1 $\times \;10^{-5}$	3	6	1404	1193	1.0672 $\times \;10^{-7}$	2.3228 $\times \;10^{-6}$	1.0000 $\times \;10^{0}$	72.152

## Data Availability

The datasets presented in this article are not readily available because of technical limitations. Requests to access the datasets should be directed to the corresponding author.

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
