# Peer review of "Domain-Separated Quantum Neural Network for Truss Structural Analysis with Mechanics-Informed Constraints"

_biomimetics, 2025, doi:10.3390/biomimetics10060407_

Round 1
Reviewer 1 Report
Comments and Suggestions for Authors
-
- The authors claim QNNs are more efficient, but they don't directly compare their results with classical neural networks or traditional FEM/Force Method in terms of computational time or resource usage. Without such comparisons, it's hard to validate their claims of superiority.
- Quantum Neural Networks are still emerging, and practical implementations on NISQ devices are challenging. The manuscript uses simulations, but there's no mention of actual quantum hardware experiments.
- Authors mention dividing the input into subdomains but don't clearly explain how the decomposition is done. Is there a risk of losing global structural information by processing domains independently?
- They use penalty and Lagrangian terms. However, the manuscript reports negative loss values in some cases (like Case 4 in Table 4), which is confusing. Loss functions are typically non-negative (e.g., Table 4, Case 4: loss = −1663×10−4−7.1663×10−4).
- the parameters like number of qubits and layers seem arbitrary in experimental section.
- The manuscript lacks detailed information on the code, data, or specific quantum simulators used.
- How do quantum properties like entanglement directly contribute to solving the problem better than classical methods?
- What is mean by, “Figure ?? compares the prediction results under no Lagraigna term and” in line 623 assign by journal
- Some issues, like inconsistent figure references (e.g., Figure ?? in page 23) and formatting inconsistencies in equations and tables. This need proofreading.
- Figure citations are inconsistent, at page no 20-make citation figure 12 and so many mistake committed by authors.
- Missing labels for axes in some figures (e.g., Figure 3, 6).
- Table 3 lists "Time (s)" but does not clarify whether this includes quantum simulation time or classical optimization.
Yes need proofreading
Author Response
Dear Reviewer,
Thank you very much for your thorough review of our manuscript. We sincerely appreciate your constructive and insightful comments.
We have carefully considered and addressed all 13 of your comments, and revised the manuscript accordingly. In particular, we have clarified the design of our study, including the input structure and domain decomposition strategy, and refined the methodological explanations to enhance clarity and completeness.
Furthermore, to improve the overall quality of the manuscript, we plan to request professional English editing service from MDPI prior to publication.
Detailed responses to each of your comments are provided in the attached Word file.
Once again, we sincerely thank you for your valuable feedback.

Reviewer 2 Report
Comments and Suggestions for Authors
The manuscript is well written and well organized and the contribution of the interest is good.
These are some comments that are better to be considered.
Consider adding a table of notations.
Consider evaluating on real world datasets.
Do a better analysis on the significance and efficiency of the work as compared with the previous approaches.
Computational complexity analysis and parameter analysis is required.
Consider an ablation study.
Author Response
Dear Reviewer,
Thank you very much for taking the time to thoroughly review our manuscript. Your constructive and insightful comments have been immensely helpful in improving the overall quality of our work.
We have carefully reviewed and addressed each of the five comments you provided and have revised the manuscript accordingly. In particular, we clarified the design of the study regarding the input structure and domain decomposition strategy, and further elaborated on the proposed methodology.
In response to your comment that the presentation of results was somewhat unclear, we thoroughly reorganized the Numerical Examples section to better convey the numerical context and significance of each result. We also strengthened the comparative analysis and interpretation across different cases. Additionally, the Conclusion section was rewritten to ensure that the conclusions are naturally drawn from the experimental results, making the connection between results and conclusions more explicit.
Finally, we will request MDPI’s professional English editing service to further improve the language quality of the manuscript.
Detailed responses to each of your comments are provided in the attached PDF file.
Once again, we sincerely thank you for your valuable feedback.

Reviewer 3 Report
Comments and Suggestions for Authors
SUMMARY
In the paper submitted for review, the authors propose a model for structural analysis of truss systems based on a quantum neural network (QNN) using a variational quantum circuit (VQC). The chosen structure reflects the way nature efficiently organizes and optimizes complex systems, which brings the study closer to the Biomimetics journal.
The reviewer notes that numerical experiments were performed on 2D and 3D truss structures to evaluate internal forces and displacements. The effects of key factors on the model performance were also analyzed.
The reviewer believes that this paper can be considered for publication, but the comments that the reviewer has raised need to be corrected. The comments are listed below.
COMMENTS
- The Abstract should include quantitative results achieved by the authors during the study.
- The Abstract should indicate in which engineering applications the developed method can be used.
- The authors are encouraged to format the article according to the IMRAD structure, which is often used in the natural and applied sciences.
- At the beginning of the Methods section, it is suggested to add a flow chart demonstrating the research steps.
- The authors are encouraged to make changes to the Introduction section by adding clear formulations of the scientific problem, scientific novelty, purpose and objectives of the study, as well as theoretical and practical significance.
- At the end of the section, in line 75, the authors talk about the R2, RMSE metrics. And further on, the RMSE metric is not used.
- The Mean Squared Error (MSE) and R2 metrics should be described, and the MSE formula should be added.
- Comments should be given on the Time (s) column in Table 3.
- The reviewer recommends checking the figures and descriptions numbers, as in some cases they do not match.
- In line 679, replace "Equation ??" with the correct value.
- The Discussion section should be highlighted, where the results obtained can be compared with the results of other researchers based on quantitative indicators.
- The practical limitations of the method should be clearly identified. For example, limitations related to performance, memory requirements, and computing resources.
The reviewer's overall conclusion on the article is that the article has certain scientific and practical prospects, but requires revision. Overall conclusion – Major Revisions.
Comments on the Quality of English LanguageEnglish editing is required.
Author Response
Dear Reviewer,
Thank you very much for taking the time to thoroughly review our manuscript. Your constructive and insightful comments have been extremely helpful in enhancing the completeness and academic contribution of our work.
We carefully reviewed and addressed each of the twelve comments you provided and accordingly revised and refined the manuscript. In particular, we clarified the rationale behind the research design, including the input structure and domain decomposition strategy, and elaborated on the proposed methodology to improve reader comprehension.
In response to your comment regarding the lack of clarity in the presentation of results, we thoroughly reorganized the Numerical Examples section. The numerical context and implications of each result have been rewritten to ensure clearer interpretation. We also strengthened the comparative performance analysis and quantitative discussions for each case. Moreover, the Conclusion section was revised to ensure that the conclusions are naturally drawn from the actual experimental results, making the connection between results and conclusions more explicit.
Additionally, we plan to improve the language quality and formatting of the entire manuscript through MDPI’s professional English editing service.
Detailed responses to each of your comments are provided in the attached Word file.
Once again, we sincerely appreciate your valuable feedback, and we hope that this revised version successfully addresses your concerns.
Thank you.

Round 2
Reviewer 2 Report
Comments and Suggestions for Authors
All concerns are addressed
Reviewer 3 Report
Comments and Suggestions for Authors
The authors answered the reviewer's questions and corrected all comments.
The corrected manuscript has become significantly better both scientifically and visually.
The reviewer has no more comments and in its current form the manuscript can be published in the journal Biomimetics.